# FOCUS: DLLMs Know How to Tame Their Compute Bound

**Kaihua Liang**[1] **Xin Tan**[2] **An Zhong**[1] **Hong Xu**[2] **Marco Canini**[1]

 https://github.com/sands-lab/FOCUS

## Abstract

Diffusion Large Language Models (**DLLMs**) offer a compelling alternative to Auto-Regressive models, but their deployment is constrained by high decoding cost. In this work, we identify a key inefficiency in DLLM decoding: while computation is parallelized over token blocks, only a small subset of tokens is decodable at each diffusion step, causing most compute to be wasted on non-decodable tokens. We further observe a strong correlation between attention-derived token importance and token-wise decoding probability. Based on this insight, we propose FOCUS, an inference system designed for DLLMs. By dynamically *focusing* computation on decodable tokens and evicting non-decodable ones on-the-fly, FOCUS increases the effective batch size, alleviating compute limitations and enabling scalable throughput. Empirical evaluations demonstrate that FOCUS achieves up to $3.52\times$ throughput improvement over the production-grade engine LMDeploy in large-batch settings, while preserving or improving generation quality across multiple benchmarks.

## 1. Introduction

Large Language Models (LLMs) have achieved significant success towards Artificial General Intelligence (Bubeck et al., 2023). However, the standard token-by-token decoding paradigm lacks parallelism and restricts global context modeling. Consequently, Diffusion Large Language Models (DLLMs) have emerged as a promising alternative, enabling simultaneous generation of multiple tokens and breaking the strict sequential dependency, which offers superior sample efficiency and modeling capabilities (Ni et al., 2025).

A remarkable milestone is LLaDA (Nie et al., 2025), which

*Table 1.* **Comparison of DLLM inference paradigms.** FOCUS further enhances the inference efficiency by predicting decodable tokens to eliminate computation redundancy.

| METHOD | PARALLEL DECODING | KV CACHE | DECODABLE PREDICTION | INFERENCE EFFICIENCY |
|---|---|---|---|---|
| LLaDA | ✗ | N/A | ✗ | ★ |
| Fast-dLLM | ✓ | Approx. | ✗ | ★★ |
| SDAR / LLaDA2.0 | ✓ | Exact | ✗ | ★★★ |
| FOCUS (Ours) | ✓ | Exact | ✓ | ★★★★ |

scales DLLMs to 8 billion (B) parameters using global bi-directional attention. Unlike sequential Auto-Regressive (AR) models, DLLMs generate text by iteratively denoising latent variables from random noise. Despite the flexibility to decode tokens in non-contiguous order, these models suffer from low inference efficiency. This is because they typically decode only a single token per diffusion step while necessitating the computation of Key-Value (KV) states for the entire sequence at every step. To mitigate this latency bottleneck, many efforts (Ma et al., 2025; Liu et al., 2025b; Wei et al., 2025; Bao et al., 2026; Chen et al., 2026) have been made to reuse the KV cache and unlock the parallel decoding capabilities of DLLMs. A representative approach is Fast-dLLM (Wu et al., 2026b), which achieves significant speedup via *confidence-based decoding* and *approximate caching*. However, these methods still require periodic KV cache recomputation, which introduces substantial latency.

To improve DLLMs further, recent works focus on the *Block-Diffusion* paradigm (Arriola et al., 2025). This approach processes a segment of tokens (block) simultaneously while treating the preceding sequence as fixed context (Figure 1). By confining the bi-directional attention within these block-wise structures, it eliminates periodic KV cache recomputation and enables exact KV cache—similar to Auto-Regressive LLMs (ARLLMs). SDAR (Cheng et al., 2025) and LLaDA2.0 (Bie et al., 2025) also leverage the high-quality pre-trained weights of ARLLMs by employing Block-Diffusion Continual Pre-Training (CPT), maintaining generation quality at scales up to 100B parameters.

However, DLLMs still face a formidable computational wall in diffusion decoding that distinguishes them from their

[1]KAUST [2]CUHK. Correspondence to: Kaihua Liang <kaihua.liang@kaust.edu.sa>, Marco Canini <marco@kaust.edu.sa>.

*Proceedings of the $43^{rd}$ International Conference on Machine Learning*, Seoul, South Korea. PMLR 306, 2026. Copyright 2026 by the author(s).

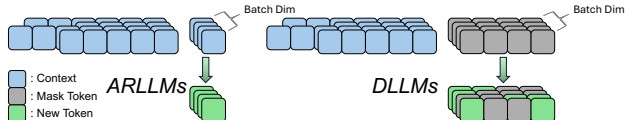

*Figure 1.* **Comparison of inference paradigms.** (Left) ARLLMs generate tokens one token at a time. (Right) DLLMs process an entire block in parallel, yet decode only a subset at every step.

*Table 2.* **Breakdown of inference computational cost per layer.** Compared to ARLLMs, DLLMs incur significantly higher FLOPs due to the block-wise query size $B$.

| Component | ARLLM | DLLM |
|---|---|---|
| Attention Projection | $8h^2$ | $B \cdot 8h^2$ |
| KV Cache Attention | $4hL$ | $B \cdot 4hL$ |
| Feed-Forward (MLP) | $4d_{ff}h$ | $B \cdot 4d_{ff}h$ |
| **Total FLOPs** | $\mathcal{O}(h^2 + hL)$ | $\boldsymbol{B \cdot \mathcal{O}(h^2 + hL)}$ |

AR counterparts (Peng et al., 2025). DLLMs must compute attention for block-wise query tokens per denoising step. This drastic increase in arithmetic intensity breaks the conventional assumption of LLM inference being *memory-bound* (Pope et al., 2023) and shifts it to a *compute-bound* regime. This reveals a critical efficiency gap: while block-wise parallelism enables exact KV cache and maximizes hardware utilization, the diffusion process redundantly recomputes the entire block each step, although only $\sim 10\%$ of the block tokens are successfully decoded. Consequently, while prior optimizations (Wu et al., 2026a; Liu et al., 2025a) effectively reduce latency in low-concurrency settings (e.g., batch size of 1), they hit a ceiling in production-scale scenarios. As batch size increases, the massive arithmetic redundancy saturates compute units, preventing DLLMs from achieving the throughput scaling essential for real-world deployment (Cheng et al., 2025; Fu et al., 2025).

To address this challenge, we aim to selectively compute decodable tokens during inference, i.e., masked tokens whose predictions satisfy the decoding criterion and can be unmasked in the current denoising step. By filtering out non-decodable tokens, we can directly reduce the *Floating Point Operations (FLOPs)* required per step, thereby significantly alleviating the compute bottleneck. While prior research on ARLLMs has exploited natural attention sparsity for historical *KV Cache Eviction* (Zhang et al., 2023; Xiao et al., 2024; Li et al., 2024; Tang et al., 2024; Cai et al., 2025), we pivot this concept to *Token Eviction within Blocks*. Specifically, we uncover a critical phenomenon in DLLMs: the *drift of token importance* within a block—manifested as incoming attention score deltas in the initial layers—is highly correlated with a token's final decoding probability.

Building on this insight, we introduce **FOCUS**, an inference system that concentrates computational resources on tokens with high decoding probabilities without any additional training. It achieves this by precisely identifying promising candidate tokens in the early layers and dynamically evicting the rest from subsequent processing. Reducing the number of processed tokens per step by approximately 65%–80%, FOCUS successfully tames the compute-bound nature of DLLM inference, yielding significant throughput gains without compromising generation quality. Table 1 compares our system against existing DLLM inference paradigms.

To summarize, our contributions are as follows:

- We identify that DLLM inference is fundamentally compute-bound. Our analysis reveals a severe inefficiency: while the models compute for full blocks, the proportion of decoded tokens is typically only $\sim 10\%$.
- To the best of our knowledge, we are the first to reveal the intrinsic relationship between attention importance and decoding probabilities in DLLMs. We demonstrate that the attention patterns in initial layers can serve as a robust predictor for token decodability.
- Based on this finding, we propose FOCUS, an inference system that mitigates compute-bound limits by evicting non-decodable tokens in a training-free manner. Against the production-grade engine LMDeploy (Zhang et al., 2025), FOCUS achieves up to **3.52×** throughput improvement, scaling throughput at large batch sizes where standard methods hit a computational wall.

## 2. Background and Motivation

**Diffusion-Based Language Models.** Diffusion-based text generation evolved from continuous embedding (Li et al., 2022) to discrete state-space formulations. D3PM (Austin et al., 2021a) introduced Markov transition matrices for noise, while SEDD (Lou et al., 2024) and MDLMs (Sahoo et al., 2024) advanced discrete diffusion through marginal probability modeling and masked absorbing states. Recent works scaled these architectures to large models. LLaDA (Nie et al., 2025) and Dream (Ye et al., 2025) reached billions of parameters, matching AR baselines through global bi-directional attention. However, despite parallel decoding potential, practical deployment remains challenging due to heavy overhead from full-sequence KV states computation at every denoising step. Fast-dLLM (Wu et al., 2026b) introduced *confidence-based decoding*, while also employing *approximate caching* alongside approaches like dLLM-Cache (Liu et al., 2025b) and dKV-Cache (Ma et al., 2025). Nevertheless, these methods trade generation quality for speed and still incur substantial periodic cache recomputation to maintain context consistency.

And so, research has increasingly converged on the Block-Diffusion paradigm (Arriola et al., 2025). Unlike ARLLMs, DLLMs process multiple query tokens concurrently at every step, as shown in Figure 1. Representative models like

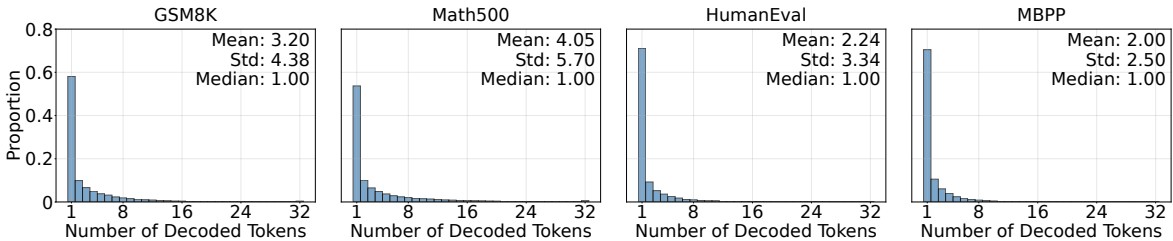

*Figure 2.* **Distribution of decoded tokens per step across benchmarks.** Using the SDAR-8B-Chat model with block size $B = 32$ and $confidence\_threshold = 0.9$, the data reveals that the mean proportion of successfully decoded tokens is typically only around 10%, indicating that $\sim 90\%$ of the block-wise computation is redundant. Please see Appendix A for more results.

SDAR (Cheng et al., 2025) and LLaDA2.0 (Bie et al., 2025) adopt a semi-AR approach: they process an entire block of tokens simultaneously while treating the preceding sequence as fixed context. While this paradigm solves the global refresh bottleneck using exact KV caches, it still redundantly computes the entire block despite decoding only a subset per step, repeating until the block is fully decoded.

**The Computational Bottleneck and Batching Limits.** The transition from ARLLMs to DLLMs fundamentally alters inference scaling properties. ARLLMs benefit from *batching*; since decoding is *memory-bound* (Pope et al., 2023), increasing batch size amortizes I/O costs, allowing throughput to scale until compute saturation. In contrast, DLLMs are strictly *compute-bound*, limiting these batching gains. Processing a block of $B$ tokens concurrently (Nie et al., 2025; Cheng et al., 2025; Bie et al., 2025) causes a surge in arithmetic intensity. As shown in Table 2 and Eq. 1, the FLOPs per layer scale linearly with the query size $Q$:

$$\text{FLOPs} \approx Q \cdot (8h^2 + 4d_{ff}h + 4hL) \qquad (1)$$

where $h$ is the hidden size, $d_{ff}$ is the intermediate MLP size, and $L$ is the context length. With a typical block size $Q = B = 32$, this load rapidly saturates GPU (Williams et al., 2009). As a result, unlike ARLLMs, DLLMs yield diminishing returns as batch sizes increase, due to a lack of idle compute cycles (Dao, 2024; Peng et al., 2025).

This saturation disguises a massive *efficiency mismatch*. While block-wise parallelism maximizes hardware utilization, Figure 2 reveals that with $B = 32$, the diffusion process redundantly recomputes the entire block to yield only 2.00–4.05 tokens on average (Cheng et al., 2025). Thus, despite full GPU utilization, $\sim$90% of the FLOPs are expended on *ineffective work*. Unlike ARLLMs' 1:1 computation-to-generation ratio, DLLMs suffer from marked structural redundancy. To restore scalability, we must alleviate the compute bound by *evicting* these redundant tokens.

## 3. Attention as Decodability Predictor

To address the computational redundancy in DLLMs, we aim to identify a low-cost metric that distinguishes "de-

codable" tokens from noise. We draw inspiration from the intrinsic *uneven attention distribution* observed in AR-LLMs. Pioneering works such as H2O (Zhang et al., 2023) and StreamingLLM (Xiao et al., 2024) demonstrated that a small subset of "heavy-hitter" and "attention-sink" tokens dominates the attention mass, a property subsequently leveraged by SnapKV (Li et al., 2024) and Quest (Tang et al., 2024) for fine-grained KV cache compression.

We pivot this insight from memory-centric cache compression to compute-centric query token eviction, directly addressing DLLM computational redundancy. By filtering out non-decodable tokens in the model's initial layers during forward pass, we can eliminate redundant computations in both the Attention and Feed-Forward Networks (FFNs) in the subsequent layers, effectively reducing their FLOPs (Kaplan et al., 2020). Adopting the *incoming attention score* as a proxy for token importance, we uncover a critical phenomenon unique to the diffusion process: this importance score exhibits an intrinsic, strong correlation with decoding probabilities, particularly in the early layers.

### 3.1. Token Importance and Layer-wise Dynamics

**Metric Definition.** To quantify token importance, we define the $j$-th token's importance $\mathcal{I}_j$ by aggregating attention weights from all query tokens in the current block $B$:

$$\mathcal{I}_j = \sum_{i,h} \text{Softmax}\left(\text{MaxPool1D}(S_{i,j}^{(h)})\right) \qquad (2)$$

where $i, j \in \{1, \ldots, B\}$ index the intra-block tokens, $h \in \{1, \ldots, H\}$ indexes the attention heads, and $S_{i,j}^{(h)} = (\mathbf{q}_i^{(h)})^\top \mathbf{k}_j^{(h)} / \sqrt{d_k}$ denotes the pre-softmax attention score between query token $i$ and key token $j$ in head $h$, with $d_k$ the per-head key dimension. Following SnapKV (Li et al., 2024), we apply MaxPool1D to robustly capture local features. As illustrated in Figure 5 (Right), this metric aggregates scores column-wise, highlighting tokens that are heavily attended to by the rest of the block.

**Empirical Observation.** We investigate how importance varies across layers during inference. As shown in Figure 3 (Top), decoded tokens naturally dominate the attention mass

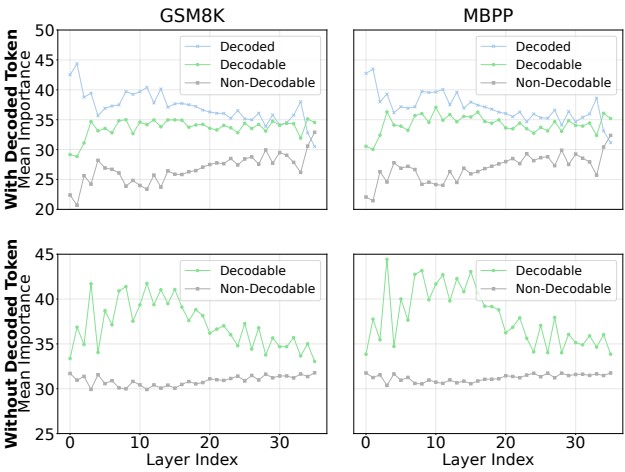

Figure 3. **Layer-wise importance.** (Top) Decoded tokens dominate. (Bottom) Filtering them reveals Decodable tokens diverge from Non-Decodable from Layer 1. Settings match Figure 2.

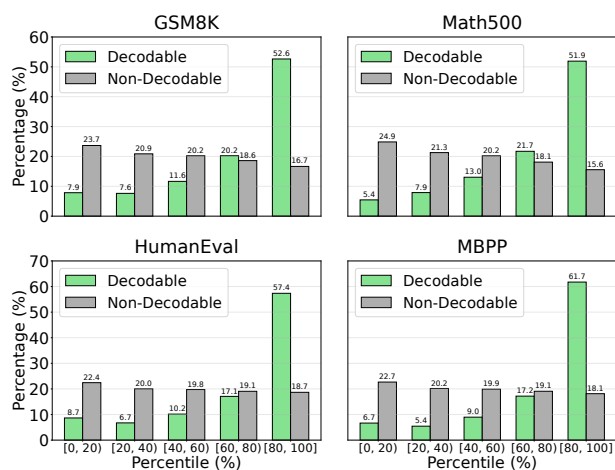

Figure 4. **Importance delta vs. decodability.** Decodable tokens (green) cluster at high deltas, distinct from Non-Decodable ones (grey). Settings match Figure 2. See Appendix B for more results.

due to the established context. To reveal the signal distribution among the candidates, we exclude these dominant tokens in Figure 3 (Bottom). This filtered view exposes a clear differentiation starting from Layer 1:

- **Layer 0:** The importance scores exhibit negligible differences between decodable and non-decodable tokens.
- **Layer 1+:** Decodable tokens gain significant attention mass, whereas non-decodable tokens are suppressed.

### 3.2. The Importance Delta Hypothesis

The divergence observed in Section 3.1 raises a key question: why does the signal emerge prominently from Layer 1?

**Signal Differentiation Mechanism.** We attribute this to the distinct roles of the early layers. The Query and Key matrices in Layer 0 are projected directly from the noisy input embeddings without cross-token interaction. Thus, we hypothesize that Layer 0's attention distribution is largely dominated by static priors or random noise, lacking discriminative context. Layer 1 instead operates on initially mixed hidden states. Analogous to the findings in bi-directional BERT (Devlin et al., 2019) model—where early layers resolve surface features (Jawahar et al., 2019)—a significant surge in attention weight at Layer 1 suggests that the token has acquired sufficient semantic coherence to distinguish itself from the others, acting as a context anchor.

**Correlation Analysis.** Thus, we propose the *importance delta*—defined as the attention score difference between the first two layers—as a robust predictor for decodability:

$$\Delta \mathcal{I}_j = \mathcal{I}_j^{(Layer1)} - \mathcal{I}_j^{(Layer0)} \qquad (3)$$

Conceptually, the subtraction acts as a *Common Mode Rejection* mechanism. By filtering non-specific Layer 0 attention

(e.g., positional priors), $\Delta \mathcal{I}$ isolates the *semantic lift* emerging in Layer 1. This two-layer delta is strategically optimal. Since Layer 1 represents the earliest instance where decodable tokens diverge from the non-decodable ones (Figure 3), identifying them at this stage maximizes the potential for compute savings across subsequent layers. To validate its correctness, we analyzed token distributions based on the percentile of their importance deltas. Figure 4 demonstrates a strong positive correlation between high percentiles and successful decoding; in contrast, tokens in lower percentiles mostly remain masked. This establishes $\Delta \mathcal{I}$ as a robust predictor, providing the theoretical basis for the early eviction mechanism detailed in Section 4. Appendix B further verifies the universality of this correlation across distinct diffusion paradigms and empirically demonstrates the superiority of the delta metric over raw attention scores.

## 4. FOCUS

We propose **FOCUS**, a DLLM inference system that selectively concentrates computational resources on tokens with the highest decoding probabilities. By filtering out non-decodable tokens, our approach increases the effective batch size to scale system throughput. As illustrated in Figure 5, the system coordinates Dynamic Budgeting (Left) and Delta Calculation (Right) to guide Token Eviction (Middle). The following sections detail our design, while the complete procedure is summarized as Algorithm 1 in Appendix C.

### 4.1. Dynamic Budgeting

To enable token eviction without compromising the parallel decoding capabilities of DLLMs, it is crucial to determine an appropriate token retention budget $K$ for each denoising step. A static budget is suboptimal, as the number of decod-

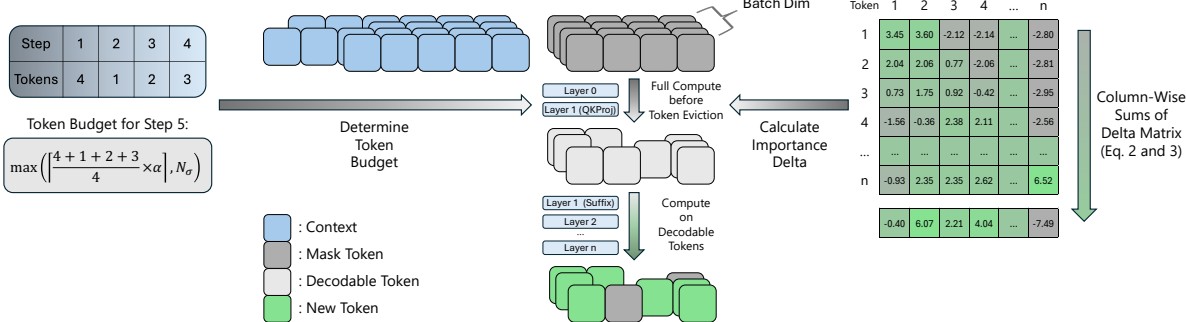

*Figure 5.* **FOCUS design overview.** The workflow centers on **(Middle)** *Token Eviction*, which performs early filtering after the Layer 1 Q/K projections (QKProj). It prioritizes decodable token candidates based on the Importance Delta ($\Delta\mathcal{I}$ in Eq. 3) from **(Right)** *Delta Calculation*, and retaining only the top candidates within the adaptive token budget determined by **(Left)** *Dynamic Budgeting*.

able tokens varies significantly across diffusion steps. We propose a dynamic budgeting mechanism based on both historical statistics and instantaneous signal strength. To handle the first denoising step where historical data is unavailable, we adopt a default of $\bar{N}_{decoded} = 1$. For all subsequent steps, the retention budget $K$ is calculated as:

$$K = \min\left(B, \max(\lceil\alpha \times \bar{N}_{decoded}^{(t)}\rceil, N_\sigma)\right) \quad (4)$$

where $\alpha > 1$ serves as the **only hyperparameter** introduced by FOCUS to control expansion aggressiveness, and $N_\sigma$ represents the count of tokens with *importance deltas* of at least the standard deviation within the block (noting that the deltas follow a zero-mean distribution):

$$N_\sigma = \sum_{j \in \mathcal{M}} \mathbb{1}\left(\Delta\mathcal{I}_j \geq \mathrm{Std}(\Delta\mathcal{I})\right) \quad (5)$$

This dual-criteria mechanism ensures robustness: the historical term $\alpha \times \overline{N}$ guarantees a safe baseline based on recent decoding yields, while the variance-based term $N_\sigma$ allows the budget to adaptively expand when the model detects a sudden surge in high-confidence tokens, preventing the filtering of valid candidates during easy-to-decode steps. Crucially, Appendix D theoretically validates that this thresholding strategy minimizes the probability of missing decodable tokens, maximizing effective parallelism.

### 4.2. Token Eviction Strategy

To translate the theoretical FLOPs reduction into realized wall-clock speedup, FOCUS implements a *token eviction* mechanism. Crucially, this mechanism operates as an early-stage filter. Guided by the strong correlation between importance delta and decoding probability established in Section 3.2, FOCUS computes the Query ($\mathbf{Q}$) and Key ($\mathbf{K}$) projections of the first two transformer layers (Layer 0 and 1) for the full block to derive the importance deltas ($\Delta\mathcal{I}$), after which the eviction logic is engaged. With the budget $K$ determined, the system first **identifies** the indices of the top-$K$ masked tokens based on these deltas. However,

to ensure generation stability, this preliminary selection is augmented by enforcing two structural constraints:

- **AR-Context Preservation:** For each selected candidate, we force the retention of its immediate predecessor. Since most DLLMs (Cheng et al., 2025; Bie et al., 2025; Fu et al., 2025) are pre-trained on AR backbones, the local attention pattern $t_i \leftarrow t_{i-1}$ remains crucial for maintaining generation coherence.

- **Placeholder Integrity:** To guarantee *positional correctness*, we retain unprocessed masked tokens preceding any selected candidate. This prevents the absence of referable KV states for these evicted positions, ensuring the attention mechanism preserves correct *relative offsets* and avoids generation corruption (e.g., repetition). The overhead is minimal, as each token is computed at most one extra time to initialize KV states, while enabling *opportunistic decoding*.

We define the final index set $\mathcal{S}$ by consolidating the top-$K$ candidates with the additional tokens required by the constraints above. Using this set $\mathcal{S}$, FOCUS performs a Gather operation on the block hidden states $\mathbf{H} \in \mathbb{R}^{B \times h}$ and their projections to produce reduced tensors (yielding $\mathbf{H}_{reduced} \in \mathbb{R}^{|\mathcal{S}| \times h}$). Subsequent layers operate exclusively on these reduced representations, while the evicted tokens serve as fixed reference KV states—guaranteed via Placeholder Integrity. Since $|\mathcal{S}| \ll B$, this linearly reduces the computational cost of matrix multiplications (Eq. 1). Crucially, as analyzed in Appendix E.5, the scheduler and eviction overhead of FOCUS is negligible, ensuring theoretical FLOPs reductions translate into wall-clock speedup.

### 4.3. Intra-Block KV Cache

Standard Block-Diffusion refreshes the KV states for the entire block at every step, incurring additional overhead. To mitigate this, we introduce an *Intra-Block KV Cache* to reuse the states of successfully decoded tokens, effectively freezing them to cease redundant updates. To implement this safely, we adapt the Delayed Cache mechanism from

*Table 3.* **Generation quality of selection strategies.** Comparison of Top, Random, and Bottom selection. **Top** consistently outperforms baselines across all retention budgets ($K \in \{2, 4, 8\}$).

| $K$ | Strategy | GSM8K | Math500 | HumanEval | MBPP | IFEval |
|---|---|---|---|---|---|---|
| | | | *Model: SDAR* | | | |
| 2 | **Top** | **90.33** | **65.40** | **71.65** | **58.56** | **61.78** |
| | Random | 40.38 | 7.50 | 4.88 | 10.32 | 47.35 |
| | Bottom | 33.44 | 4.40 | 1.22 | 1.56 | 43.85 |
| 4 | **Top** | **90.60** | **65.90** | **70.12** | **56.03** | **60.88** |
| | Random | 61.04 | 27.40 | 15.55 | 20.43 | 48.14 |
| | Bottom | 51.82 | 10.80 | 3.97 | 5.45 | 43.82 |
| 8 | **Top** | **89.84** | **66.10** | **68.90** | **57.01** | **60.26** |
| | Random | 78.55 | 46.90 | 43.29 | 37.36 | 53.46 |
| | Bottom | 60.39 | 21.60 | 9.15 | 13.62 | 44.35 |
| | | | *Model: LLaDA2.0* | | | |
| 2 | **Top** | **89.99** | **60.80** | **84.76** | **80.16** | **82.74** |
| | Random | 51.71 | 26.40 | 16.77 | 22.18 | 70.47 |
| | Bottom | 12.67 | 6.70 | 7.32 | 14.01 | 62.20 |
| 4 | **Top** | **91.02** | **61.50** | **85.68** | **81.13** | **83.25** |
| | Random | 76.20 | 45.30 | 36.59 | 39.30 | 73.34 |
| | Bottom | 40.52 | 20.90 | 11.59 | 20.82 | 67.25 |
| 8 | **Top** | **90.22** | **63.80** | **83.54** | **79.58** | **81.91** |
| | Random | 84.58 | 57.30 | 64.64 | 59.92 | 75.12 |
| | Bottom | 67.18 | 40.70 | 23.17 | 33.08 | 68.54 |

*Table 4.* **Generation quality across thresholds and $\alpha$ on SDAR.** Comparison of Baseline (B) and FOCUS (F) across confidence thresholds ($Conf \in \{0.9, 0.8, 0.7\}$) and expansion factors ($\alpha \in \{1.2, 1.5, 1.8\}$). The highest scores for each threshold are in **bold**.

| Conf. | M | $\alpha$ | GSM8K | Math500 | HumanEval | MBPP | IFEval |
|---|---|---|---|---|---|---|---|
| 0.9 | B | - | 89.20 | **64.70** | 69.82 | 56.81 | 57.45 |
| | F | 1.2 | 89.84 | 64.60 | **72.56** | 58.17 | **60.97** |
| | F | 1.5 | 90.15 | 64.30 | 69.51 | **61.09** | 60.87 |
| | F | 1.8 | **90.75** | 63.80 | 71.34 | 58.95 | 60.11 |
| 0.8 | B | - | 87.57 | 59.30 | 65.85 | 50.78 | 58.69 |
| | F | 1.2 | **90.33** | **64.10** | 67.68 | 58.56 | 59.91 |
| | F | 1.5 | 89.73 | 62.20 | 69.21 | **59.73** | 60.97 |
| | F | 1.8 | 89.39 | 62.10 | **69.82** | 56.81 | **61.14** |
| 0.7 | B | - | 84.65 | 54.60 | 59.76 | 50.58 | 58.47 |
| | F | 1.2 | 89.24 | 62.00 | **68.29** | **56.81** | 61.91 |
| | F | 1.5 | **89.31** | **62.20** | 64.33 | 55.25 | **62.26** |
| | F | 1.8 | 88.03 | 59.80 | 64.33 | 53.89 | 60.08 |

dKV-Cache (Ma et al., 2025). This approach postpones committing tokens to the cache until they stabilize—specifically, after being decoded and forwarded one additional time.

However, direct application compromises generation quality due to the violation of local dependency chains. In CPT-based architectures (Cheng et al., 2025; Bie et al., 2025; Fu et al., 2025), the generation of a token $t_{i+1}$ heavily relies on the attention features from its immediate predecessor $t_i$. Prematurely freezing the KV states of $t_i$ before $t_{i+1}$ has been decoded introduces noise into this critical dependency. Therefore, we introduce a *Neighbor-Aware Stability Criterion*: we delay the caching of $t_i$'s KV states until both $t_i$ *and* its right neighbor $t_{i+1}$ are successfully decoded. This ensures that the local context window is fully stabilized before the KV states are finalized, preventing error propagation across the long-term generation process.

## 5. Evaluation

In this section, we evaluate FOCUS by first verifying that it preserves the model's generation quality, and then demonstrating its substantial improvement in inference efficiency.

**Implementation.** Building on **LMDeploy** (Zhang et al., 2025), which officially supports the Block-Diffusion paradigm, FOCUS integrates scheduler logic and custom *Triton kernels* (Tillet et al., 2019) for irregular memory access, spanning 4,000+ lines. Appendix E provides details.

**Datasets.** We evaluate the generation quality and inference efficiency of FOCUS as follows:

- **Generation Quality Benchmarks.** We assess the system's capabilities across three primary domains:
  - *Mathematical Reasoning*: **GSM8K** (Cobbe et al., 2021) and **Math500** (Hendrycks et al., 2021).
  - *Code Generation*: **HumanEval** (Chen et al., 2021) and **MBPP** (Austin et al., 2021b).
  - *Instruction Following*: **IFEval** (Zhou et al., 2023).

- **Inference Throughput Benchmarks.** We assess the system's efficiency using larger-scale datasets:
  - *Real-World Chat*: **ShareGPT** (ShareGPT, 2023) and **WildChat** (Zhao et al., 2024).
  - *Complex Logic*: **MATH** (Hendrycks et al., 2021), which benchmarks the interplay between semantic accuracy and decoding speed (Cheng et al., 2025).

**Models.** We utilize the following state-of-the-art (SOTA) DLLMs to evaluate the effectiveness of FOCUS:

- **SDAR-8B-Chat** (Cheng et al., 2025): Our default evaluation model, which natively supports diverse block sizes for efficiency benchmarking.

- **LLaDA2.0-mini** (Bie et al., 2025): A Mixture-of-Experts (MoE) DLLM with 16B total and 1.4B active parameters, demonstrating SOTA capabilities.

For brevity, we refer to SDAR-8B-Chat as **SDAR** and LLaDA2.0-mini as **LLaDA2.0** throughout the experiments.

**Hardware.** We conduct all experiments on a single NVIDIA **A100-SXM4-80GB** GPU.

**Default Settings.** Following (Wu et al., 2026b; Fu et al., 2025; Bie et al., 2025), we set default block size $B$ to 32 and confidence threshold to 0.9. See Appendix F.1 for details.

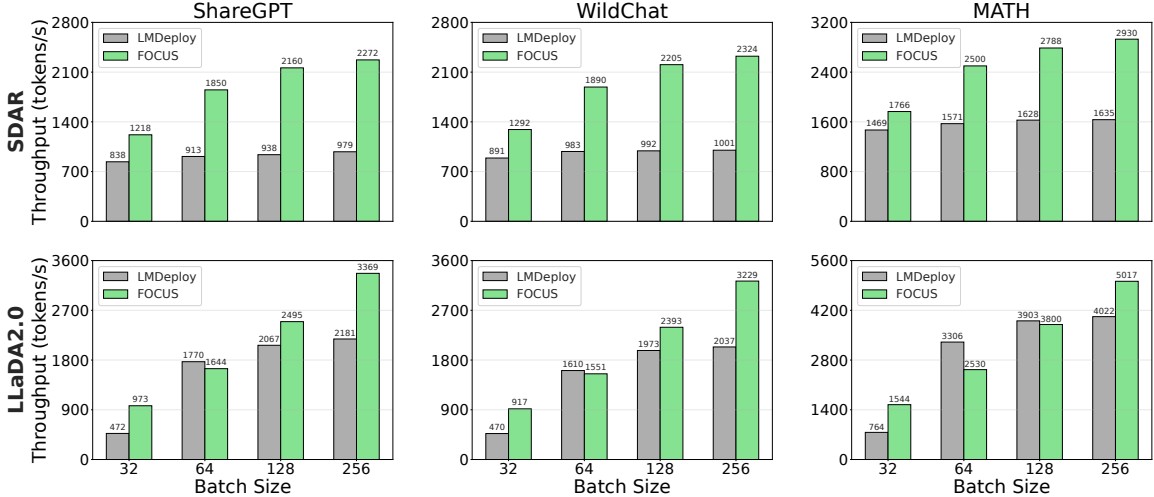

*Figure 6.* **Throughput scaling.** FOCUS achieves up to **2.32× throughput** improvement over the LMDeploy baseline. FOCUS breaks the computational wall, enabling sustained throughput growth at larger batch sizes for both SDAR and LLaDA2.0 models.

### 5.1. Validation of Importance-Based Selection

To verify that the *Importance Delta* is a robust predictor for token selection, we compare the generation quality using fixed Top/Random/Bottom-$K$ to replace selection strategy in FOCUS. Table 3 presents the generation quality when fixing the retention budget to $K \in \{2, 4, 8\}$ as FOCUS at least retains 2 tokens with $\alpha > 1$ in Eq. 4. The quality hierarchy of *Top > Random > Bottom* confirms that the *Importance Delta* is a reliable and accurate metric for decodable token selection. As the budget $K$ increases, the generation quality of *Random* and *Bottom* strategies improves, as a larger $K$ raises the chance of including the correct decodable tokens.

In contrast, the *Top* strategy remains robust across all budget settings. Since the inference process operates as a Markov chain (Cheng et al., 2025), selecting incorrect tokens as decodable candidates—as frequently occurs in *Random* and especially *Bottom* strategies—introduces errors that accumulate during denoising, severely degrading generation quality.

### 5.2. Overall Generation Quality

We investigate the sensitivity of FOCUS to varying expansion factors ($\alpha$) and confidence thresholds. As presented in Table 4, FOCUS consistently matches or outperforms the corresponding Baseline across all settings. Crucially, FOCUS demonstrates superior **robustness** at lower confidence thresholds. As the threshold relaxes from 0.9 to 0.7, the Baseline suffers from significant performance degradation (e.g., a drop from 64.70 to 54.60 on Math500). This decline occurs because lower thresholds inadvertently admit "high-confidence but incorrect" tokens (false positives), which pollute the context for subsequent diffusion steps.

In contrast, FOCUS effectively mitigates this degradation.

Even at a relaxed threshold of 0.7, FOCUS ($\alpha = 1.5$) maintains high generation quality (e.g., 62.20 on Math500), significantly surpassing the Baseline. One mechanistic interpretation is that importance-based eviction may suppress noisy candidates (i.e., non-decodable tokens) that the confidence model fails to catch; we treat these quality trends as indirect evidence rather than a strict causal analysis, and leave controlled causal validation to future work. Based on this robustness, we adopt $\alpha = 1.5$ and Conf $= 0.8$ as the default configuration, as it yields an average score of 68.37 across benchmarks—outperforming even the conservative Baseline at Conf $= 0.9$ (67.60).

### 5.3. End-to-End Throughput

We evaluate throughput (tokens/s) across batch sizes on ShareGPT, WildChat, and MATH datasets, by randomly sampling 5,000 requests per dataset and setting the maximum generation length to 2048 tokens. We utilize the SOTA **LMDeploy** (Zhang et al., 2025) engine as our baseline. Crucially, it has leveraged production-grade optimizations, including *Continuous Batching* (Yu et al., 2022), *PagedAttention* (Kwon et al., 2023), and *FlashAttention* (Dao, 2024), ensuring comparison against a highly efficient system.

**Significant Throughput Improvement.** As illustrated in Figure 6, the baseline LMDeploy exhibits a distinct throughput plateau as batch size increases. For instance, on SDAR-ShareGPT, throughput saturates at approximately 900 tokens/s across batch sizes 32 to 256. In contrast, FOCUS effectively restores scalability. By evicting non-decodable tokens, our approach reduces the effective FLOPs per step, allowing throughput to grow with batch size. Quantitatively, as shown in Table 5, FOCUS drastically reduces the computational redundancy ratio ($N_{processed}/N_{decoded}$), dropping

*Table 5.* **Computational redundancy (Layer 2+).** Average ratio of processed to decoded tokens ($N_{processed}/N_{decoded}$); lower is more efficient. FOCUS significantly reduces redundancy.

| Model | Dataset | Baseline | FOCUS | Reduction |
|---|---|---|---|---|
| SDAR | ShareGPT | 15.02 | **3.12** | 79.23% |
| | WildChat | 14.83 | **3.05** | 79.43% |
| | MATH | 7.45 | **2.69** | 63.89% |
| LLaDA2.0 | ShareGPT | 19.73 | **4.19** | 78.76% |
| | WildChat | 21.47 | **4.30** | 79.97% |
| | MATH | 10.13 | **3.04** | 69.99% |

*Table 6.* **Ablation of intra-block KV cache on SDAR.** Neighbor-Aware Stability Criterion (DC+) prevents quality degradation from standard Delayed Cache (DC). The highest scores are in **bold**.

| Method | GSM8K | Math500 | HumanEval | MBPP | IFEval |
|---|---|---|---|---|---|
| Baseline | 89.20 | **64.70** | **69.82** | 56.81 | 57.45 |
| DC | 84.92 | 59.20 | 64.33 | 49.81 | 57.94 |
| DC+ | 88.02 | 63.70 | 69.21 | 52.14 | 55.15 |
| FOCUS | **89.73** | 62.20 | 69.21 | **59.73** | **60.97** |

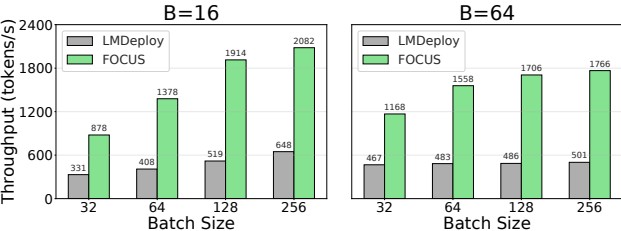

*Figure 7.* **Throughput scaling across block sizes ($B$).** On ShareGPT and SDAR, FOCUS achieves up to **3.52× speedup** at $B = 64$, where the computational burden is heaviest.

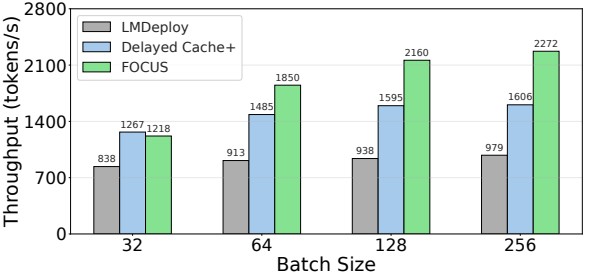

*Figure 8.* **Throughput breakdown on ShareGPT and SDAR.** While Intra-Block KV cache aids efficiency, the full FOCUS system provides substantial speedup without compromising quality.

from 15.02 to 3.12 on ShareGPT (a ∼79% reduction). This significant drop indicates that FOCUS effectively bridges the efficiency gap, shifting the diffusion decoding paradigm much closer to the ideal 1:1 computation-to-generation ratio inherent in ARLLMs. Consequently, in *Real-World Chat* scenarios, FOCUS achieves peak throughputs of **2,272 tokens/s** on ShareGPT and **2,324 tokens/s** on WildChat (Batch Size=256), representing a **2.32× speedup** over LMDeploy. Notably, this realized speedup verifies that the system overhead is negligible. Our profiling shows that scheduler adaptation and token eviction consume only ∼1% of step latency (analyzed in Appendix E.5), ensuring that theoretical FLOPs reductions translate directly into wall-clock performance.

**Impact of Block Size.** We further analyze the sensitivity of throughput to the diffusion block size $B$, as shown in Figure 7. For both smaller ($B = 16$) and larger ($B = 64$) block sizes, FOCUS consistently delivers substantial throughput gains. The advantage is particularly pronounced at $B = 64$, where the computational redundancy intensifies; here, FOCUS achieves a peak speedup of **3.52×**, demonstrating it can tame the computational burden of large-block diffusion.

**Robustness Across Architectures.** On LLaDA2.0 (MoE), FOCUS maintains its advantage across datasets. For instance, on ShareGPT, it boosts throughput from 2,181 to 3,369 tokens/s at a batch size of 256. The relative speedup is more moderate compared to dense SDAR, as LLaDA2.0 utilizes only **1.4B** active parameters, which inherently reduces the FLOPs per token. This effect is particularly noticeable in the *Complex Logic* benchmark MATH. Here, shorter prompt lengths limit the computational burden of attention, leaving

less room for optimization. Moreover, at batch size of 64, a slight regression occurs due to the disablement of multi-loop optimization (Appendix E.3.4), where scheduler overhead outweighs FLOPs reduction in this long reasoning scenario. Nevertheless, FOCUS still delivers substantial acceleration at higher loads, verifying its effectiveness even on inherently efficient architectures and challenging reasoning tasks.

### 5.4. Ablation Study

**Neighbor-Aware Stability Criterion.** Table 6 highlights the critical sensitivity of block-diffusion to KV cache consistency. With the confidence threshold fixed at 0.9 for both variants, the standard Delayed Cache (DC) incurs severe quality degradation by freezing states prematurely. This violates the intrinsic $t_i \leftarrow t_{i+1}$ dependency chain required by CPT-adapted models. Our Neighbor-Aware strategy (DC+) successfully restores stability. Crucially, however, the complete FOCUS system achieves generation quality that surpasses even the original Baseline. This reaffirms that FOCUS acts as a proactive safety filter, purging high-confidence but incorrect tokens that escape the threshold.

**System Efficiency Gains.** Figure 8 deconstructs the relative contributions to inference speedup. Although intra-block KV caching (DC+) provides a baseline efficiency boost with slight quality degradation compared to the LMDeploy baseline, FOCUS's dynamic token eviction provides the substantial gain without compromising generation integrity.

## 6. Discussion

**Related Work.** FOCUS shares an insight with attention-guided inference methods: attention patterns expose inference-time redundancy. In ARLLMs, prior works use uneven attention mass to evict historical KV cache entries and shorten the effective cache length (Zhang et al., 2023; Xiao et al., 2024; Li et al., 2024; Tang et al., 2024; Cai et al., 2025); in DLLMs, Sparse-dLLM (Song et al., 2026) similarly evicts unimportant cache entries, while d$^2$Cache (Jiang et al., 2026) adaptively caches and reuses KV states across diffusion steps. FOCUS differs in operating target and regime: we find that the early-layer *importance delta* correlates with token decodability and use it for query-token eviction within the current diffusion block, thereby reducing Attention and FFN computation in the remaining layers. Recently, LLaDA2.1 (Bie et al., 2026) proposes token editing, also orthogonal to FOCUS; future work could use early-layer signals to predict which positions require recomputation for either editing or decoding.

**System Support and Remaining Gap.** FOCUS adds scheduling logic on top of LMDeploy's DLLM stack (Zhang et al., 2025), but this overhead is small compared with the saved computation (Appendix E.5). Nevertheless, DLLM inference still lacks the mature system support of ARLLM inference stacks, leaving a larger gap to highly optimized AR-LLM systems. Appendix F.2 additionally reports batch-size-1 throughput, where scheduling overhead is less amortized. Improving DLLM-native scheduling and GPU-resident state management remains important future work.

## 7. Conclusion

We identify a fundamental inefficiency in DLLM inference, where standard block-wise processing wastes ∼90% of compute on non-decodable tokens. To tame this, we introduce the **FOCUS** system, which dynamically evicts non-decodable tokens early in the forward pass. Crucially, our analysis reveals that *token decodability* is not random but predictable via early-layer attention patterns. Evaluations confirm efficiency gains while maintaining generation quality: at standard $B = 32$, it achieves up to **2.32×** throughput improvement, scaling to **3.52×** with larger $B = 64$. Beyond establishing a robust training-free baseline, FOCUS opens a promising research direction: shifting from blind, redundant computation to proactive, predictive decoding. We hope this work inspires future research into more sophisticated decodability predictors, further unlocking the efficiency potential of diffusion-based language models.

## Acknowledgements

We are thankful to the anonymous reviewers for their thoughtful comments and suggestions that helped improve our paper.

## Impact Statement

This paper presents work whose goal is to advance the field of Machine Learning. There are many potential societal consequences of our work, none which we feel must be specifically highlighted here.

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

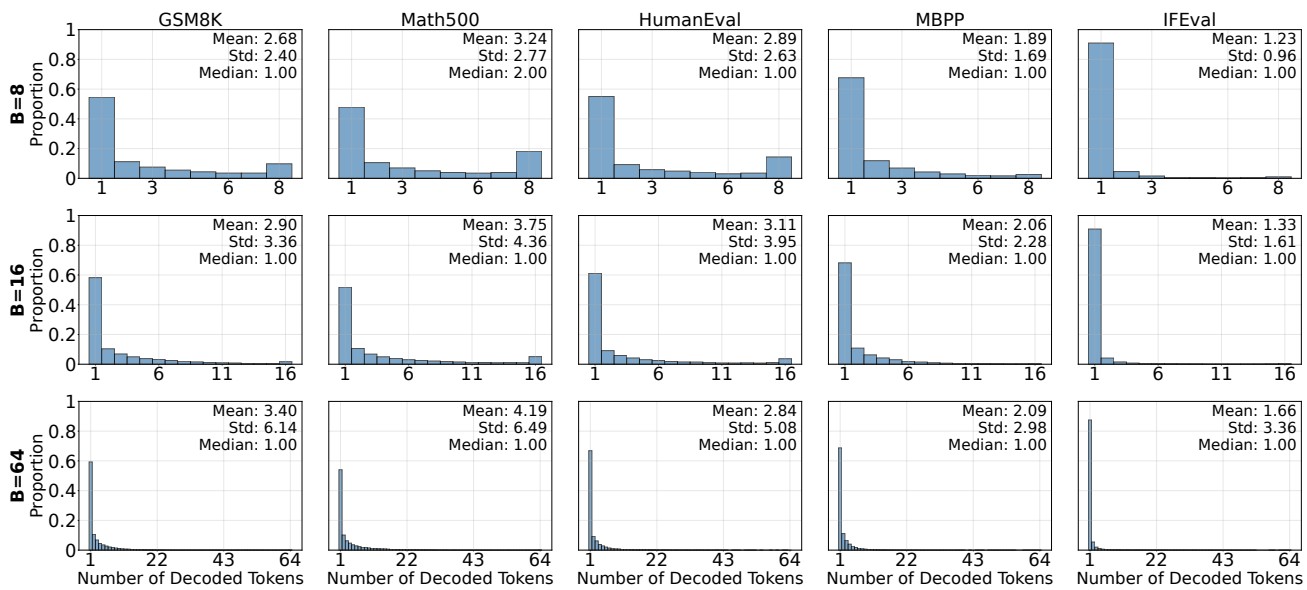

*Figure 9.* **Decoded token distribution for SDAR across block sizes $B \in \{8, 16, 64\}$.** Histograms show the number of decoded tokens per step across all benchmarks. Rows correspond to different block sizes, while columns represent different datasets.

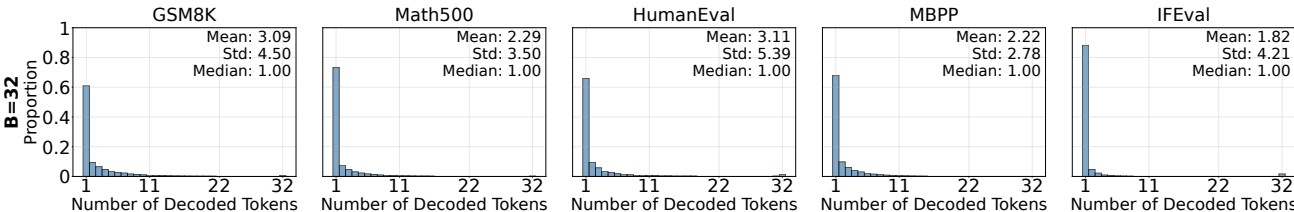

*Figure 10.* **Decoded token distribution for LLaDA2.0-mini ($B = 32$).** Histograms showing the number of decoded tokens per step across all evaluated benchmarks.

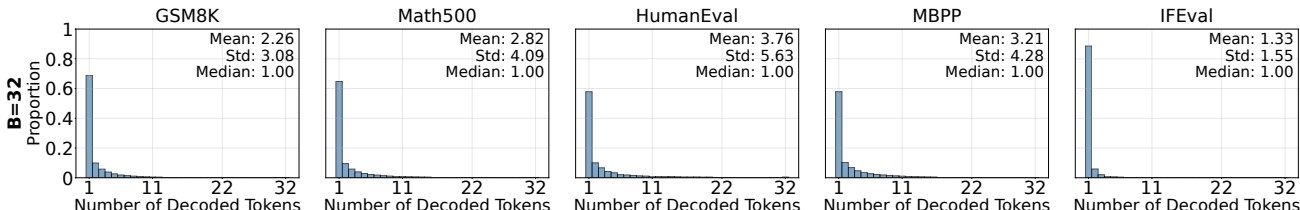

*Figure 11.* **Decoded token distribution for LLaDA-8B-Instruct ($B = 32$).** Histograms showing the number of decoded tokens per step across all evaluated benchmarks.

## A. Extended Analysis of Decoded Token Statistics

We conduct an extended analysis of the decoding statistics across three representative DLLMs, including the **SDAR** family (Cheng et al., 2025), **LLaDA 2.0-mini** (Bie et al., 2025), and **LLaDA-8B-Instruct** (Nie et al., 2025), where the latter utilizes **Fast-dLLM** (Wu et al., 2026b) as the inference framework. To investigate the impact of block size, we configure *SDAR-8B-Chat* model with varying block sizes $B \in \{8, 16, 64\}$.

Consistent with the analysis presented in Section 2, we utilize the standard benchmarks for mathematical reasoning, namely **GSM8K** (Cobbe et al., 2021) and **Math500** (Hendrycks et al., 2021), as well as **HumanEval** (Chen et al., 2021) and **MBPP** (Austin et al., 2021b) for code generation tasks. Furthermore, to provide assessment of instruction-following capabilities, we additionally incorporate **IFEval** (Zhou et al., 2023) into this extended analysis.

Figures 9, 10 and 11 present the histograms of decoded tokens per step. The results consistently demonstrate that, regardless

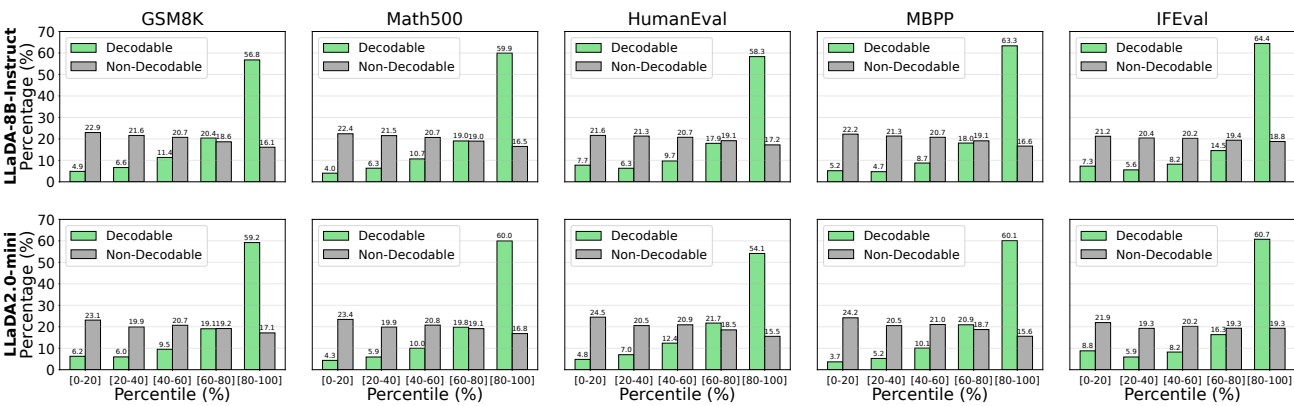

Figure 12. **Importance delta vs. decodability across all benchmarks.** Columns represent individual datasets, while rows correspond to LLaDA-8B-Instruct (Full-Diffusion) and LLaDA2.0-mini (Block-Diffusion), respectively.

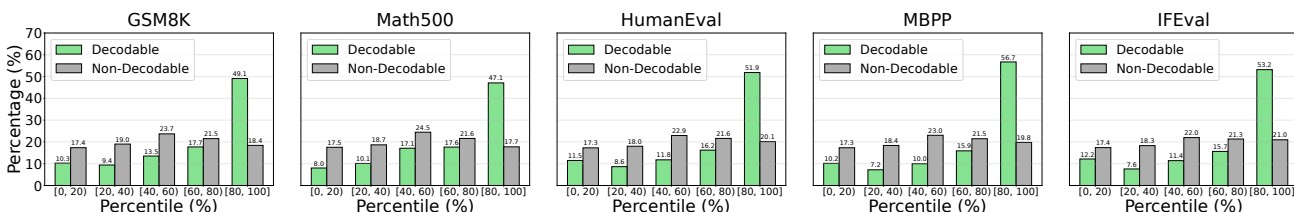

Figure 13. **Layer 1 importance vs. decodability across all benchmarks.** Instead of using *importance delta*, we use the raw importance scores at Layer 1 directly in this figure. Settings match Figure 4.

of the model architecture or the configured block size, only a small fraction of tokens within a block are successfully decoded in each denoising step. This inefficiency is particularly pronounced in challenging tasks such as **IFEval**, where the models frequently decode only a single token per step. These findings confirm that massive computational redundancy is an intrinsic characteristic of current block-diffusion decoding paradigms.

## B. Extended Correlation Analysis

In this section, we provide an extended empirical validation of the correlation between the *Importance Delta* and token decodability, broadening the core insight discussed in Section 4. Figure 12 illustrates the conditional probability of token decoding as a function of the importance score percentiles across all evaluated datasets. The top row displays the results for **LLaDA 2.0-mini** (Bie et al., 2025), representing the *Block-Diffusion* paradigm, while the bottom row presents **LLaDA-8B-Instruct** (Nie et al., 2025), representing the *Full-Diffusion* paradigm.

We observe a consistent and sharp polarization across all benchmarks: successfully decoded tokens are overwhelmingly concentrated within the highest percentiles of the importance delta distribution, whereas those in lower percentiles exhibit a negligible decoding probability. Crucially, this pattern holds true not only for the Block-Diffusion paradigm of LLaDA 2.0 but also for the global attention mechanism of the Full-Diffusion LLaDA-8B-Instruct. This cross-paradigm consistency suggests that the strong correlation between *importance delta* and *decoding probability* is a **universal characteristic intrinsic** to MDLMs (Sahoo et al., 2024). These results further justify the design of FOCUS, confirming that the *importance delta* serves as a robust, architecture-agnostic signal for identifying decodable tokens. This implies that the applicability of FOCUS extends beyond the currently SOTA Block-Diffusion paradigm, paving the way for its adaptation to accelerate a wider spectrum of diffusion-based LLMs.

To further substantiate the design choice of the *Importance Delta* ($\Delta \mathcal{I} = \mathcal{I}^{(Layer1)} - \mathcal{I}^{(Layer0)}$), we compare it against using the raw importance scores from Layer 1 alone. As illustrated in Figure 13, relying solely on Layer 1 scores results in a marked reduction in discriminative power compared to the delta metric shown in Figure 4. For instance, on the **HumanEval** benchmark, the concentration of decodable tokens in the highest percentile bin ([80, 100]) drops from **57.4%** (using $\Delta \mathcal{I}$) to **51.9%** (using raw Layer 1 scores). Furthermore, distinct from the delta metric where non-decodable tokens are effectively suppressed in higher percentiles, the distribution in Figure 13 reveals that non-decodable tokens do not decrease as sharply

---

**Algorithm 1** FOCUS Inference Step

---

1: **Input:** Masked Block $\mathbf{X} \in \mathbb{R}^{B \times d}$, Context $\mathbf{C}$, Cumulative Mean $\bar{N}_{\text{decoded}}^{(t)}$
2: **Hyperparameters:** Expansion factor $\alpha > 1$, Block Size $B$
3: *// Phase 1: Initial Layer Forward & Metrics*
4: *// Compute Layer 0 fully and Layer 1 projections (Query/Key) for importance*
5: $\mathbf{H}^{(0)}, \mathbf{H}_{\text{proj}}^{(1)} \leftarrow \text{Forward}(\mathbf{X}, \mathbf{C}, \text{Layers} = 0, 1_{\text{proj}})$
6: $\Delta \mathcal{I} \leftarrow \text{Importance}(\mathbf{H}_{\text{proj}}^{(1)}) - \text{Importance}(\mathbf{H}^{(0)})$ {Eq. 3}
7: $N_\sigma = \sum_{j \in \mathcal{M}} \mathbb{1}(\Delta \mathcal{I}_j \geq \text{Std}(\Delta \mathcal{I}))$ {Eq. 5}
8: *// Phase 2: Dynamic Budgeting*
9: **if** $\bar{N}_{\text{decoded}}^{(t)}$ is uninitialized (Step 1) **then**
10:     $K_{hist} \leftarrow \lceil \alpha \times 1 \rceil$
11: **else**
12:     $K_{hist} \leftarrow \lceil \alpha \times \bar{N}_{\text{decoded}}^{(t)} \rceil$
13: **end if**
14: $K \leftarrow \min(B, \max(K_{hist}, N_\sigma))$ {Eq. 4}
15: *// Phase 3: Decodable Token Selection & Execution*
16: $\mathcal{S} \leftarrow \text{TopK\_Indices}(\Delta \mathcal{I}, K)$
17: $\mathcal{S} \leftarrow \mathcal{S} \cup \{i - 1 \mid i \in \mathcal{S}\}$ {AR-Context Preservation}
18: $\mathcal{S} \leftarrow \mathcal{S} \cup \{j \mid j < \max(\mathcal{S}), j \text{ is masked}\}$ {Placeholder Integrity}
19: *// Gather hidden states and finish Layer 1 computation on reduced set*
20: $\mathbf{H}_{reduced} \leftarrow \text{Gather}(\mathbf{H}_{\text{proj}}^{(1)}, \mathcal{S})$
21: $\mathbf{O} \leftarrow \text{Forward}(\mathbf{H}_{reduced}, \mathbf{C}, \text{Layers} = 1_{\text{suffix}} \ldots L)$
22: $\mathbf{Y}_{decoded} \leftarrow \text{Decode\_and\_Verify}(\mathbf{O})$
23: Update $\bar{N}_{\text{decoded}}^{(t+1)}$ with $|\mathbf{Y}_{decoded}|$ as a cumulative arithmetic mean
24: Update KV Cache with Neighbor-Aware Stability
25: **Return** $\mathbf{Y}_{decoded}$

---

with rising percentiles. This indicates that raw attention scores contain static noise (e.g., positional priors) that obscures the signal, validating the necessity of the subtraction. This empirical observation aligns with our theoretical analysis in Appendix D, which demonstrates that the delta metric minimizes the error probability by transforming the heavy-tailed raw attention into a statistically separable distribution.

Finally, we address the computational implication of this dual-layer calculation. As detailed in the overhead analysis in Appendix E.5, the cost of computing the *importance Delta* is negligible. Since the importance calculation operates on the block-wise attention matrix with complexity $\mathcal{O}(B^2)$—where the block size $B$ is typically small (e.g., 32)—the additional FLOPs required to compute Layer 0 scores and the subsequent difference are virtually non-existent compared to the $\mathcal{O}(d_{ff}h)$ complexity of the model's dense layers. Thus, the *Importance Delta* provides superior signal clarity with minimal overhead.

## C. Pseudocode of FOCUS

We provide the detailed inference procedure of FOCUS in Algorithm 1. The algorithm takes the masked block sequence and context as input. In the first phase (Lines 3-6), it performs a partial forward pass to compute the *importance delta* ($\Delta \mathcal{I}$) and the variance-based count $N_\sigma$. In the second phase (Lines 7-12), it dynamically calculates the token budget $K$ using both cumulative arithmetic decoding statistics ($\bar{N}_{\text{decoded}}^{(t)}$) and the instantaneous signal ($N_\sigma$). Finally, in the third phase (Lines 13-20), it selects decodable tokens, enforces structural constraints (*AR-Context Preservation* and *Placeholder Integrity* in Section 4.2), and executes the forward pass on the reduced token set. We fill the KV states into the KV cache before the token eviction in Layer 1.

## D. Theoretical Analysis of Eviction Safety

In this section, we provide a theoretical justification for the safety of our token eviction mechanism. We model the token selection process in FOCUS as a binary hypothesis testing problem within the framework of Signal Detection Theory (Green

& Swets, 1966). Our goal is to derive an upper bound on the probability of erroneously evicting a valid **decodable token** (Type II Error), which represents a loss in **parallel decoding potential** for the current diffusion step.

### D.1. Probabilistic Model

Let $x$ denote a token in the sequence, and let $Y \in \{0, 1\}$ be a latent variable indicating its decodability, where $Y = 1$ represents a decodable token and $Y = 0$ represents a non-decodable one. We utilize the Importance Delta, $\Delta\mathcal{I}$, as the decision statistic.

Based on our empirical observations in Section 3 and prior studies on attention sparsity (Zhang et al., 2023; Li et al., 2024), we model the distribution of $\Delta\mathcal{I}$ under the two hypotheses as a first-order approximation:

- **Null Hypothesis ($H_0$, Non-decodable):** The importance delta reflects non-specific background fluctuations. We assume the non-decodable tokens follow a zero-mean Gaussian distribution:

$$\Delta\mathcal{I} \mid (Y = 0) \sim \mathcal{N}(0, \sigma^2) \tag{6}$$

  where $\sigma^2$ is the variance of the importance deltas for non-decodable tokens.

- **Alternative Hypothesis ($H_1$, Decodable):** The token acquires semantic significance at Layer 1, resulting in a positive shift in importance. We model this as:

$$\Delta\mathcal{I} \mid (Y = 1) \sim \mathcal{N}(\mu, \sigma^2) \tag{7}$$

  where $\mu > 0$ represents the strength of the decodability signal. The Signal-to-Noise Ratio (SNR) is defined as $\gamma = \frac{\mu}{\sigma}$.

### D.2. Error Bound Derivation

FOCUS employs a dynamic thresholding mechanism. For the variance-based criterion ($N_\sigma$ in Eq. 5), the effective decision threshold is set at one standard deviation above the mean of the non-decodable distribution. Thus, the eviction condition is:

$$\text{Evict} \iff \Delta\mathcal{I} < \tau, \quad \text{where } \tau = \sigma \tag{8}$$

A critical error occurs when a valid decodable token ($Y = 1$) fails to meet this threshold and is evicted (False Negative). The probability of this error, $P_{err}$, is given by:

$$P_{err} = P(\Delta\mathcal{I} < \sigma \mid Y = 1) \tag{9}$$

Standardizing the variable under $H_1$, let $Z = \frac{\Delta\mathcal{I} - \mu}{\sigma} \sim \mathcal{N}(0, 1)$. The condition $\Delta\mathcal{I} < \sigma$ transforms to:

$$\sigma Z + \mu < \sigma \tag{10}$$

$$Z < 1 - \frac{\mu}{\sigma} \tag{11}$$

$$Z < 1 - \gamma \tag{12}$$

Thus, the error probability is expressed using the Cumulative Distribution Function (CDF) of the standard normal distribution, $\Phi(\cdot)$:

$$P_{err} = \Phi(1 - \gamma) = \Phi(-(\gamma - 1)) = Q(\gamma - 1) \tag{13}$$

where $Q(x) = 1 - \Phi(x)$ is the Q-function. Using Mills' ratio for the Gaussian tail, $Q(x) \leq \frac{\phi(x)}{x}$ for $x > 0$, where $\phi(x) = \frac{1}{\sqrt{2\pi}} \exp(-x^2/2)$ is the standard normal density. Assuming the signal is sufficiently strong ($\gamma > 1$), we obtain the sharper tail bound:

**Proposition D.1** (Mills-Ratio Eviction Error Bound). *For a signal-to-noise ratio $\gamma > 1$, the probability of erroneously evicting a decodable token is bounded by:*

$$P_{err} \leq \frac{1}{(\gamma - 1)\sqrt{2\pi}} \exp\left(-\frac{(\gamma - 1)^2}{2}\right). \tag{14}$$

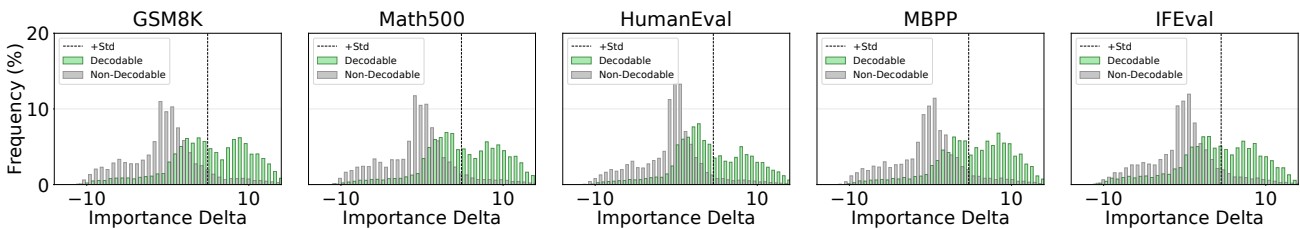

*Figure 14.* **Distribution of importance delta ($\Delta\mathcal{I}$) across benchmarks.** Decodable tokens (green) consistently exhibit higher $\Delta\mathcal{I}$ values than non-decodable tokens (grey). The dashed line indicates the $\sigma$ threshold (+Std) used for dynamic budgeting. Settings match Figure 2.

### D.3. Interpretation and Robustness

The derived bound indicates that the probability of losing a valid token follows the standard Gaussian tail decay, with an exponential factor in the square of the effective SNR and a $\frac{1}{\gamma-1}$ prefactor from Mills' ratio.

- **High SNR Regime:** As observed in Figure 3, decodable tokens typically exhibit high Importance Delta values (large $\mu$). Although the threshold is set near the **non-decodable baseline** ($\tau \approx \sigma$), the "heavy-hitter" nature of attention implies a large Signal-to-Noise Ratio ($\gamma \gg 1$). Consequently, the effective safety margin ($k = \gamma - 1$) is substantial, driving $P_{err}$ rapidly towards zero.

- **Systematic Safeguard:** Even in rare cases where the instantaneous signal $\mu$ is weak, FOCUS's dual-criteria budgeting (Eq. 4) compensates by incorporating the cumulative arithmetic mean of decoded tokens ($\alpha \times \bar{N}^{(t)}_{\text{decoded}}$), effectively increasing the retention budget beyond the count derived solely from $\sigma$.

**Remark on Distributional Assumption and Conservative Estimation.** While we acknowledge that *raw* attention scores typically exhibit heavy-tailed characteristics (e.g., Zipfian) (Zhang et al., 2023), it is important to distinguish the proposed **Importance Delta ($\Delta\mathcal{I}$)** from raw attention importance. As a differential metric defined by the subtraction of scores between layers ($\Delta\mathcal{I} = \mathcal{I}^{(Layer1)} - \mathcal{I}^{(Layer0)}$), the operation empirically *tends to reduce* the extreme positive skewness.

Furthermore, due to the nature of the Softmax operation (where the sum of weights is fixed), a surge in importance for decodable tokens often induces a slight **negative drift** in the distribution of non-decodable tokens. However, in our theoretical model (Eq. 7), we conservatively approximate the non-decodable distribution as centered at zero ($\mu_{non-decodable} = 0$). This assumption renders our derived error bound (Eq. 14) strictly **conservative**: in reality, the negative shift of the non-decodable distribution further widens the separation between decodable and non-decodable tokens ($\mu_{decodable} - \mu_{non-decodable} > \mu$), meaning the actual probability of erroneous eviction is likely even lower than our theoretical upper bound. As substantiated by the empirical distributions in Figure 14, modeling the non-decodable background as a zero-mean Gaussian serves as a tractable, safe, and robust proxy for this analysis.

**Conclusion.** This analysis confirms that selecting tokens based on statistical deviations ($\Delta\mathcal{I} > \sigma$, Eq. 4 and 5) is a robust strategy. Crucially, these findings further validate the rationale for computing $\Delta\mathcal{I}$ rather than utilizing raw attention magnitudes directly. Unlike raw attention, where the heavy-tailed distribution blurs the boundary between signal and noise, $\Delta\mathcal{I}$ offers a statistically separable distribution. Consequently, FOCUS minimizes the risk of missed decoding opportunities (which preserves parallel throughput) while maximizing token eviction efficiency.

## E. System Implementation and Optimization Details

### E.1. Software Framework and Architecture

FOCUS is architected as a high-performance extension to LMDeploy (Zhang et al., 2025), a state-of-the-art inference engine optimized for production environments. As outlined in the system overview (see Figure 5 in Section 4), the implementation comprises over 4,000 lines of code (LOC), spanning custom scheduler logic in Python and highly optimized compute kernels in OpenAI Triton (Tillet et al., 2019). The system is designed for modularity, integrating seamlessly with existing production features such as *Continuous Batching* (Yu et al., 2022) and *PagedAttention* (Kwon et al., 2023).

The architecture adopts a hybrid execution model to balance flexibility and performance. High-level control flow and the

*Dynamic Budgeting* mechanism (Section 4.1) are managed by the host-side Python interpreter, ensuring rapid prototyping and dynamic logic handling. Conversely, performance-critical operations—specifically token importance computation, sparse gathering, and state compaction—are implemented as JIT-compiled Triton kernels. Although written in a Python-based DSL, these kernels are compiled directly into efficient GPU machine code, allowing FOCUS to intervene in the decoding loop with minimal runtime overhead.

### E.2. Custom Kernel Optimizations

The non-contiguous token eviction strategy introduced in Section 4.2 generates irregular memory access patterns that are inefficient for standard dense kernels. To address this, we implement a suite of specialized kernels optimized for sparse operations.

#### E.2.1. IMPORTANCE COMPUTATION KERNEL

We implement `focus_importance_ragged_kernel` to compute the token importance metric defined in Section 3.1. To handle the variable sequence lengths within a batch efficiently, the kernel operates on ragged tensors. It computes the aggregate attention score for each token by summing weights from all query tokens in the current block, strictly following Eq. 2. We fuse the smoothing operation directly into the reduction kernel to minimize memory traffic:

$$\mathcal{I}_j = \sum_{i=1}^{B} \sum_{h=1}^{H} \text{Softmax}\left(\text{MaxPool1D}\left(S_{i,j}^{(h)}, k = 3\right)\right) \tag{15}$$

where $S_{i,j}^{(h)}$ denotes the pre-softmax attention score between query $i$ and key $j$ in head $h$. The kernel utilizes shared memory workspaces to perform atomic accumulations of softmax-normalized weights, ensuring high memory bandwidth utilization.

#### E.2.2. TOKEN SELECTION AND CONSTRAINT ENFORCEMENT

The selection logic is encapsulated in `focus_select_enforce_ragged_kernel`. This kernel is responsible for identifying the subset of tokens to retain, $\mathcal{S}$, based on the *Importance Delta* $\Delta\mathcal{I}$ established in Section 3.2. Crucially, it enforces the structural constraints proposed in Section 4.2 in a single pass to maintain generation stability:

- **AR-Context Preservation:** For every candidate token $t_i$, the kernel enforces the inclusion of $t_{i-1}$ to preserve the local autoregressive dependency chain required by CPT-based models.

- **Placeholder Integrity:** To maintain correct relative positional embeddings, all unprocessed masked tokens preceding any selected candidate are strictly retained (effectively retaining tokens up to the maximum index in the selected set).

- **Minimum Retention Guarantee:** The kernel enforces a lower bound on the retained set size ($|\mathcal{S}| \geq 1$) to prevent context collapse.

The kernel applies the dynamic thresholding mechanism (Eq. 5), selecting tokens where $\Delta\mathcal{I} > \mu + \sigma$ (where $\mu, \sigma$ are the block-wise mean and standard deviation). This statistical filtering makes the selection robust to varying noise levels across different layers and diffusion steps.

#### E.2.3. FUSED STATE COMPACTION

To translate the theoretical FLOPs reduction (analyzed in Section 2) into wall-clock speedup, we implement `focus_compact_states_kernel`. This fused kernel eliminates the overhead of multiple small memory copies by compacting all relevant inference tensors—KV states, hidden states, Residual connections, Rotary Embeddings (RoPE), and cache indices—into a dense, reduced format in a single launch. The kernel utilizes a prefix-sum (scan) algorithm to calculate destination indices in parallel, using device-side atomics to synchronize writes without host intervention.

#### E.2.4. RAGGED PAGEDATTENTION AND SPARSE CACHE MANAGEMENT

Standard attention kernels assume contiguous query blocks, an assumption broken by FOCUS's token eviction. To support efficient computation on the reduced token set $\mathcal{S}$, we implement two critical kernels:

- **Ragged PagedAttention:** We implement `_fwd_grouped_split_ragged_kernel` (wrapped as `ragged_paged_attention_fwd`) to handle non-contiguous queries. Unlike standard FlashAttention which operates on fixed blocks, this kernel utilizes an indirect indexing map, `tile_to_seq`, to map physical CUDA thread blocks to logical variable-length token sequences. This enables the GPU to saturate compute throughput even when the number of retained tokens varies significantly across the batch.

- **Sparse KV Cache Filling:** The `_fill_kv_cache_sparse_kernel` is designed to write non-contiguous KV states into the PagedAttention memory pool. It uses pre-computed `ProcessingIndices` derived during the selection phase to scatter KV states to their correct physical block offsets, ensuring that the global context remains consistent despite the sparse execution of the current step.

### E.3. Scheduler and Memory Management

E.3.1. FOCUS SCHEDULER STATE MANAGEMENT

The FOCUS scheduler maintains a multi-level state tracking system to coordinate token eviction across the host-side scheduler and GPU compute units. This design ensures consistency while minimizing synchronization overhead through asynchronous data transfers and pinned memory pools.

**Per-Sequence State Tracking** To manage block-wise decoding progress, each sequence maintains a `FocusState` object that tracks the rightmost processed position within the current block:

$$\text{rightmost\_processed}^{(t)} = \max\left(\text{rightmost\_processed}^{(t-1)}, p_{\text{current}}\right) \tag{16}$$

where $p_{\text{current}}$ is the position of the token decoded at step $t$. This state is updated via the scheduler after each decode step.

Additionally, each sequence tracks historical decoding statistics to support the dynamic budgeting mechanism. We compute the cumulative arithmetic mean of decoded tokens as follows:

$$\text{token\_sum}^{(t)} = \text{token\_sum}^{(t-1)} + n_{\text{decoded}}^{(t)} \tag{17}$$

$$\text{total\_steps}^{(t)} = \text{total\_steps}^{(t-1)} + 1 \tag{18}$$

$$\bar{N}_{\text{decoded}}^{(t)} = \frac{\text{token\_sum}^{(t)}}{\text{total\_steps}^{(t)}} \tag{19}$$

where $n_{\text{decoded}}^{(t)}$ is the count of newly unmasked tokens at step $t$.

**Delayed Cache State Coordination** The `DelayedCacheState` maintains a boolean tensor, `uncached_positions`, corresponding to the block length. This state governs the write-permission to the KV cache:

- **Warmup Phase:** Initially, all positions are marked as uncached to ensure full block processing.

- **Neighbor-Aware Updates:** Post-decoding, positions are marked as cached only when both the token and its right neighbor are unmasked, enforcing the stability criterion (Section 4.3).

- **Block Reset:** Upon full block caching, the state resets to process the subsequent block.

**Runtime View and Asynchronous Synchronization** The `FocusRuntimeView` dataclass encapsulates the device-side state required for kernel execution, including the `block_progress` tensor (tracking processed positions) and the `processing_mask` (identifying evictable tokens).

To minimize host-device synchronization overhead, the scheduler employs a double-buffered pinned memory pool:

1. **Preparation Phase:** The scheduler aggregates per-sequence metadata (local indices, processed pointers, and decoding averages).

2. **Host-Side Packing:** Values are packed into pre-allocated pinned buffers to eliminate allocation overhead during the forward pass.

3. **Asynchronous Transfer:** Data is copied to the device via non-blocking operations, recording CUDA events for synchronization.

4. **GPU Execution:** Kernels atomically update the `block_progress` state and compute importance metrics without host intervention.

5. **Async Readback:** The scheduler waits on the recorded event before reading back updated progress, ensuring consistency without blocking the computation stream.

**State Persistence and Reset**   When a block completes processing (all tokens become cached), the scheduler resets per-sequence states:

- `FocusState.rightmost_processed` is reset to $-1$.

- `DelayedCacheState.uncached_positions` is reset to all `True`.

- The warmup flag is set to trigger full processing for the next block.

This reset mechanism ensures that each block starts with a clean state while preserving the cumulative arithmetic mean statistics across blocks for stable budgeting.

### E.3.2. DYNAMIC BUDGETING

The budgeting logic, implemented in `focus_compute_targets`, determines the retention budget $K$ dynamically to balance throughput and quality. The budget is derived via Eq. 4:

$$K = \min\left(B, \max\left(\lceil \alpha \times \bar{N}_{\text{decoded}}^{(t)} \rceil, N_\sigma\right)\right) \tag{20}$$

where $\alpha$ is the expansion factor and $\bar{N}_{\text{decoded}}^{(t)}$ represents the cumulative arithmetic mean of decoding yield. Historical statistics are tracked per-sequence in GPU registers to minimize latency.

### E.3.3. ASYNCHRONOUS CACHE UPDATES

A major challenge in block eviction is managing the KV cache for tokens that are computed but not yet fully stable. We address this with the **Neighbor-Aware Stability Criterion** detailed in Section 4.3, implemented in `focus_processing_view_kernel`.

- **Stability Tracking:** The kernel maintains a `block_progress` pointer for each sequence. A token's KV state is only committed to the global PagedAttention cache once its right-side neighbor has been successfully decoded.

- **Metadata Management:** To avoid blocking the GPU, metadata updates (e.g., `cu_seqlens`, history lengths) are prepared in pinned host memory and transferred asynchronously to the device via CUDA streams. This overlaps scheduler overhead entirely with the compute-heavy attention operations.

### E.3.4. INTERACTION WITH MULTI-LOOP OPTIMIZATION

The LMDeploy engine implements a **multi-loop optimization** strategy (specifically in `engine_loop.py`) to improve GPU utilization. By executing multiple forward passes for a single batch before returning control to the host scheduler, it amortizes the kernel launch and synchronization overheads.

However, enabling FOCUS (via `enable_delayed_cache = True`) necessitates serial per-step state management, requiring us to disable this optimization (forcing `num_loops = 1`). This constraint is a fundamental requirement to guarantee generation correctness under three critical dependencies:

- **State Synchronization:** The scheduler must update the per-sequence `FocusState` and `DelayedCacheState` after *every* token generation. Skipping this update to run multiple loops would cause the eviction logic to desynchronize from the actual decoding progress, leading to invalid KV cache reuse.

- **Dynamic Budgeting Dependency:** The retention budget $K^{(t+1)}$ strictly relies on the cumulative arithmetic mean statistics updated at step $t$ (Eq. 4). A multi-loop execution implies deciding budgets for future steps before the current step's statistics are finalized, which would force the use of stale data and degrade selection accuracy.

- **Causal Cache Consistency:** The *Neighbor-Aware Stability Criterion* (Section 4.3) imposes a causal barrier: the KV states of token $t_i$ cannot be committed to the global cache until token $t_{i+1}$ is verified. Multi-loop execution would risk prematurely committing unstable states before the verification logic triggers.

**Net Performance Gain:** Despite disabling it—which inherently places FOCUS at an engineering disadvantage against the full LMDeploy baseline with this optimization—our approach delivers substantial throughput improvements. As empirically demonstrated in Section 5.3 (Figure 6 and Figure 7), FOCUS achieves up to **3.52× throughput** improvement. This confirms that the massive reduction in arithmetic intensity achieved by our token eviction strategy drastically outweighs the scheduling overhead incurred.

**Future Optimization Path:** We propose **In-Loop State Update** as a direct solution to reconcile FOCUS with this multi-loop optimization. By injecting the state update logic—implemented as a lightweight Triton kernel—directly into the existing forward loop, the system can perform *immediate, in-place updates* of decoding metadata on the GPU after each forward pass. Consequently, the eviction logic for the next iteration can access the fresh state instantly without needing to yield control back to the host scheduler. It is worth noting, however, that while this optimization sometimes improve throughput, it inherently coarsens the scheduling granularity, deviating from the strict iteration-level control flow of standard *continuous batching* mechanism (Yu et al., 2022).

### E.4. Hybrid CUDA Graph Execution

Standard CUDA Graphs require static tensor shapes, which conflicts with FOCUS's dynamic token eviction. We resolve this by partitioning the forward pass into two distinct phases:

#### E.4.1. PHASE 1: EAGER PREFIX DYNAMIC EXECUTION

The initial layers (Layer 0 and 1) execute in eager mode to accommodate the variable workload. This phase handles the highly dynamic operations discussed in Section 4.2: importance calculation, token selection, and state compaction. By keeping this phase outside the graph, we allow the tensor shapes to vary freely based on the semantic content of the input. Code entry point: `SDARModel.forward_focus_prefix`.

#### E.4.2. PHASE 2: GRAPH-CAPTURED SUFFIX WITH HYBRID BUCKETIZATION

The post-eviction layers (Layer 2 through $L$) operate on compacted tensors. While CUDA Graphs are essential for eliminating kernel launch overheads in this phase, standard capture strategies are ill-suited for the dynamic nature of token eviction.

**Hybrid Bucketization Strategy:** Unlike LMDeploy's standard strategy, which bucketizes based on the *running batch size* (padding to the nearest registered batch count), FOCUS operates on compacted tensors where the effective dimension is the *total retained token count* ($N_{retained}$). Since $N_{retained}$ fluctuates significantly per step due to dynamic eviction, we implement a hybrid strategy to balance graph reuse against padding overhead:

- **Fine-Grained Power-of-2 (for $N_{retained} < 256$):** When the number of retained tokens is small, the relative overhead of padding is significant. To minimize FLOPs waste in highly sparse scenarios (e.g., math reasoning or IFEval), we follow the classic power-of-2 alignment:

$$N_{graph} \in \{1, 2, 4, 8, 16, 32, 64, 128\} \tag{21}$$

- **Coarse-Grained Linear Stepping (for $N_{retained} \geq 256$):** For larger retention counts, the relative impact of padding diminishes. To prevent an explosion in the number of captured graphs—which would increase warm-up time and GPU

memory fragmentation—we switch to a linear stepping strategy with a stride of 256:

$$N_{graph} = \left\lceil \frac{N_{retained}}{256} \right\rceil \times 256 \tag{22}$$

During inference, the scheduler selects the smallest pre-captured graph such that $N_{graph} \geq N_{retained}$ and pads the input tensors accordingly. This hybrid approach allows FOCUS to maintain high compute efficiency during sparse execution while ensuring robust memory management during dense execution phases.

### E.4.3. GRAPH WARMUP STRATEGY

During the warmup phase, we explicitly capture graphs for reduced batch sizes to accommodate the effects of eviction. The capture logic is adjusted as follows:

```
if self.inputs_strategy.enable_focus:
    # Capture graphs for partial batch sizes
    # to optimize for post-eviction execution
    focus_cap = (max_batches + 1) // 2
    capture_batch_sizes = sorted({
        min(bs, focus_cap) for bs in capture_batch_sizes
    })
```

### E.5. Overhead Analysis

We profile the system on an NVIDIA **A100-SXM4-80GB** GPU to quantify the overhead introduced by FOCUS components. Our profiling demonstrates that the **total overhead**—encompassing metric computation, token selection, and synchronization—accounts for only approximately $1\%$ of the step latency:

- **Metric Computation:** The importance score metric's $\mathcal{O}(B^2)$ complexity is orders of magnitude smaller than the LLM backbone's $\mathcal{O}(d_{ff}h)$ (where $d_{ff} > h \gg B$).

- **Selection Latency:** The sorting and gathering operations are highly efficient with customized Triton kernels and contribute minimally to the critical path.

- **Synchronization:** By using asynchronous CUDA events for host-device communication, the scheduler overhead is effectively hidden.

This minimal overhead ensures that the theoretical reduction in arithmetic intensity translates directly into end-to-end throughput gains.

## F. Supplementary Experimental Results

### F.1. Detailed Experimental Setup

**Evaluation Frameworks.** Following (Cheng et al., 2025), we conduct the evaluation of generation quality using the *Hugging Face Transformers Library* backend with the **OpenCompass** framework (OpenCompass Contributors, 2023). For inference efficiency assessments, we utilize the **LMDeploy** engine (Zhang et al., 2025). To ensure reliability, we report the average results over two runs.

**Parameters.** Given that DLLM block sizes are typically moderate, we consistently set the kernel size of the MaxPool1D operation in Eq. 2 to 3 consistent with the configuration defined in Eq. 15. For throughput benchmarks, the maximum generation length is set to 2048 tokens, whereas for all other experiments, it is fixed at 1024 tokens.

**Datasets.** For IFEval (Zhou et al., 2023), we report the mean of prompt-level and instruction-level strict accuracy.

*Table 7.* **Batch-size-1 throughput.** End-to-end throughput (tokens/s) under the same benchmarking protocol as Section 5.3, but with batch size fixed to 1. Larger values are better.

| Model | Dataset | LMDeploy | FOCUS |
|---|---|---|---|
| SDAR | ShareGPT | 57.5 | **63.3** |
| | WildChat | 57.2 | **63.2** |
| | MATH | **122.5** | 89.6 |
| LLaDA2.0 | ShareGPT | **45.5** | 43.2 |
| | WildChat | **42.6** | 41.7 |
| | MATH | **90.7** | 70.9 |

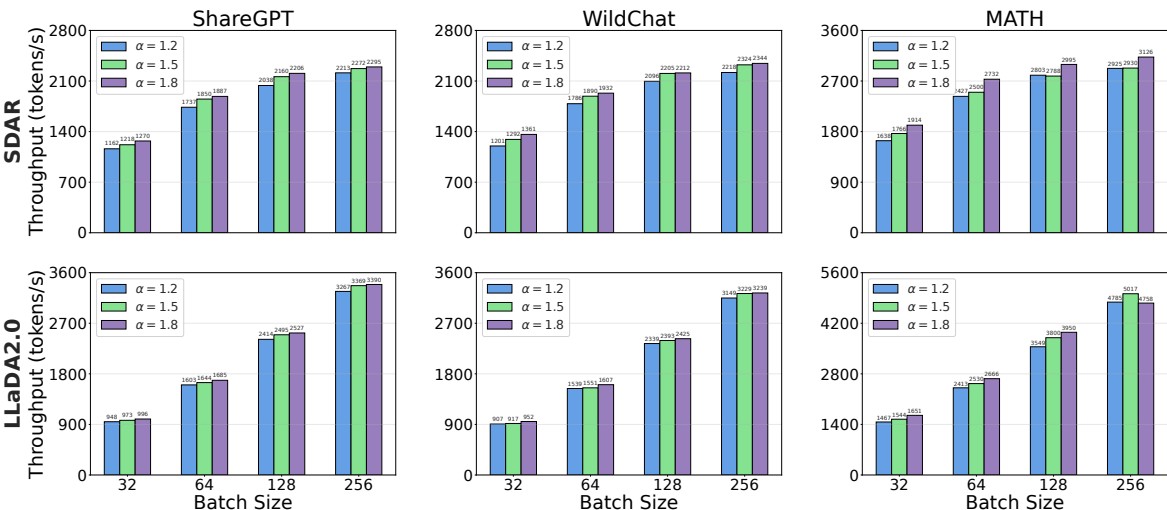

*Figure 15.* **Throughput sensitivity to expansion factor $\alpha$.** Throughput remains strong across $\alpha$ values. Larger $\alpha$ improves throughput by retaining candidates at small batches, while the trend is not monotonic because expansion can introduce redundant computation.

**Throughput Benchmarks.** Given that semantic context affects the decoding speed of DLLMs (Cheng et al., 2025), we integrate a mechanism within the benchmarking script to detect and terminate requests exhibiting excessive repetition. Although we have implemented CUDA Graph support within our system, we observed that the overhead associated with data copying to the buffer frequently counteracted the potential gains, thereby reducing overall efficiency for both the LMDeploy baseline and FOCUS. Consequently, CUDA Graph was disabled for all throughput experiments.

### F.2. Batch-Size-1 Throughput

Table 7 reports the low-concurrency batch-size-1 setting. This regime differs from the high-concurrency, compute-bound setting targeted by FOCUS. At larger batch sizes, evicting non-decodable tokens reduces the computation per decoding step and restores throughput scaling; at batch size 1, however, the amount of computation is small and scheduler overhead is more visible. As a result, FOCUS is slightly below LMDeploy on average in this setting, although it still improves throughput on the SDAR chat benchmarks. In practical scene, FOCUS can therefore be enabled adaptively: sufficiently large running batches use token eviction, while very small batches fall back to the original LMDeploy decoding path.

### F.3. Throughput Sensitivity to Expansion Factor

We further evaluate how the expansion factor $\alpha$ affects end-to-end throughput under the same benchmarking protocol as Section 5.3. As shown in Figure 15, larger $\alpha$ generally improves throughput by retaining more potential tokens, especially at small batch sizes. The trend is not monotonic in every setting, since aggressive expansion can introduce redundant computation and partially offset the benefit of exposing additional decodable candidates. Together with the generation-quality results in Table 4, these measurements support our choice of $\alpha = 1.5$ as a balanced default. More broadly, FOCUS maintains strong throughput across a wide range of $\alpha$ values, indicating that its efficiency gains are not sensitive to a narrowly tuned expansion factor.

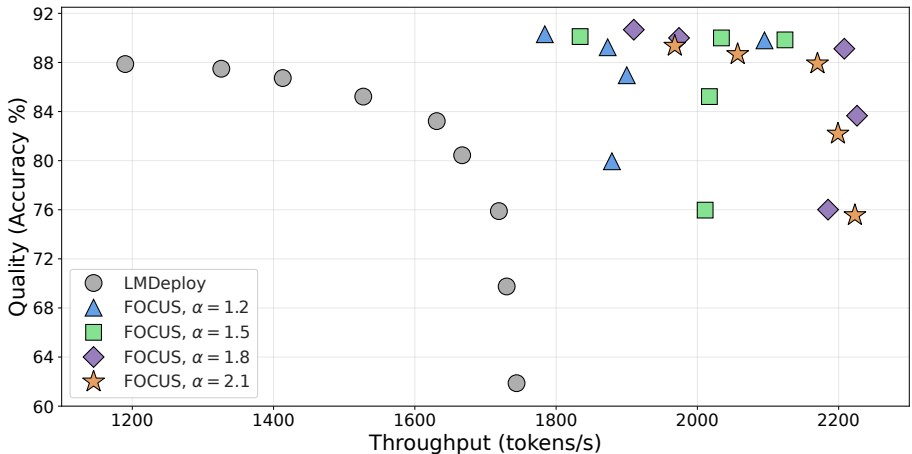

*Figure 16.* **Joint quality–throughput tradeoff on GSM8K.** We evaluate SDAR-8B-Chat at $B = 32$ and batch size 256. LMDeploy uses confidence thresholds Conf. $\in \{0.5, 0.55, \ldots, 0.9\}$, while FOCUS uses Conf. $\in \{0.5, 0.6, 0.7, 0.8, 0.9\}$ and $\alpha \in \{1.2, 1.5, 1.8, 2.1\}$. FOCUS shifts the quality–throughput frontier outward and preserves strong accuracy across substantially higher throughput regimes.

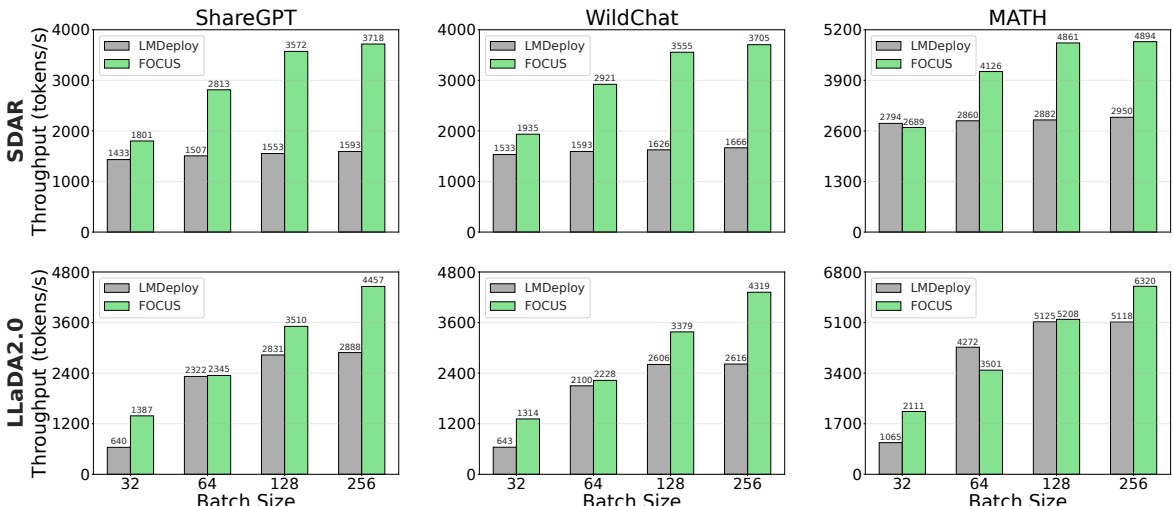

*Figure 17.* **Throughput scaling on H100.** Under the same throughput setting as Section 5.3 and Figure 6, FOCUS maintains its advantage over LMDeploy on an NVIDIA H100-80G-PCIe GPU, achieving up to **2.33× throughput** improvement.

## F.4. Quality–Throughput Tradeoff on GSM8K

We additionally measure the joint quality–throughput frontier on GSM8K (Cobbe et al., 2021) to isolate how confidence thresholding and FOCUS's expansion factor jointly affect system behavior. GSM8K provides enough examples to reveal granular changes in accuracy while still exercising long-form mathematical reasoning. For the LMDeploy baseline, we vary the confidence threshold from 0.5 to 0.9; for FOCUS, we sweep the same broad confidence range together with $\alpha \in \{1.2, 1.5, 1.8, 2.1\}$. As shown in Figure 16, FOCUS consistently dominates the LMDeploy frontier over the tested operating points, reaching substantially higher throughput while maintaining comparable or better accuracy.

The frontier also illustrates why simply relaxing the decoding criterion is insufficient for efficient DLLM inference. Because the denoising process evolves as a Markov chain, incorrectly committing noisy or non-decodable tokens can perturb subsequent decoding states. This effect appears as a near-vertical region on the right side of Figure 16: aggressive confidence relaxation, or overly large expansion in FOCUS, causes a large quality drop with little additional throughput gain and occasionally a slight slowdown. In contrast, the balanced operating region around $\alpha = 1.5$ and Conf. $= 0.8$ improves throughput while avoiding this unstable regime, reinforcing the default configuration used in our main experiments.

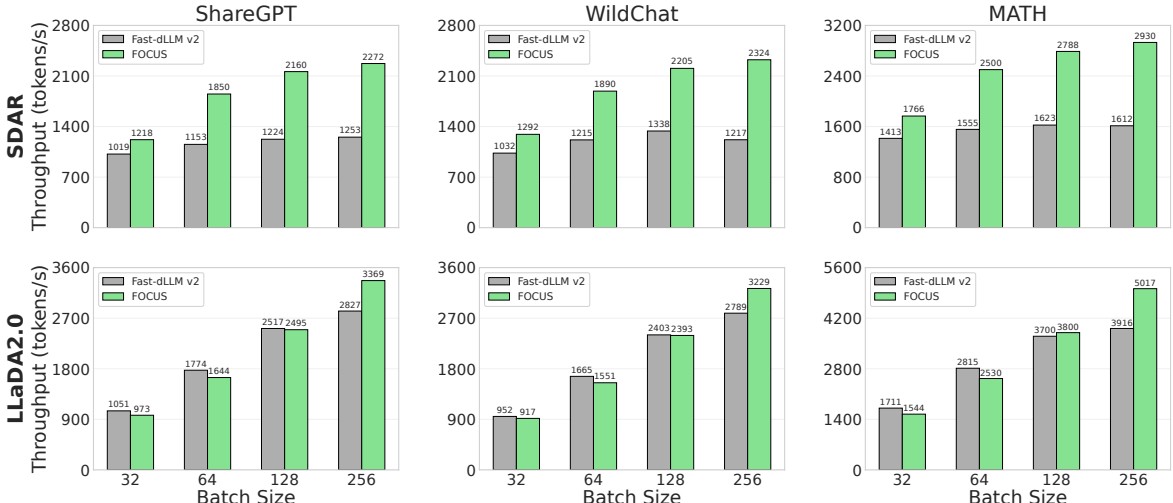

Figure 18. **Throughput comparison between Fast-dLLM v2 7B and FOCUS-SDAR.** FOCUS significantly outperforms Fast-dLLM v2, achieving up to **8.35× the throughput** on ShareGPT dataset.

Figure 19. **Engine-controlled throughput comparison with Fast-dLLM v2.** We implement Fast-dLLM v2 inside LMDeploy using the settings in Section 5.3. Fast-dLLM v2 improves over vanilla LMDeploy, but FOCUS scales better and achieves up to **1.91× throughput**.

## F.5. Throughput on Hopper GPU

To assess whether the efficiency gains of FOCUS generalize to newer GPU architectures, we additionally run throughput experiments on an NVIDIA **H100-80G-PCIe** GPU under the same setting as Section 5.3 and Figure 6. Specifically, we evaluate SDAR and LLaDA2.0 on ShareGPT, WildChat, and MATH with batch sizes 32, 64, 128, and 256. As shown in Figure 17, FOCUS continues to scale more effectively than the LMDeploy baseline on the Hopper architecture, delivering up to **2.33× throughput** improvement. These results indicate that the advantage of FOCUS is not specific to the A100-SXM4 platform used in the main experiments; instead, reducing computational redundancy through token eviction remains beneficial on newer high-end GPU hardware.

## F.6. Comparison with Fast-dLLM v2

To provide a comprehensive evaluation of inference efficiency, we benchmark **FOCUS** with SDAR-8B-Chat against the official implementation of **Fast-dLLM v2** 7B (Wu et al., 2026a) across the same datasets employed in Section 5.3 (ShareGPT (ShareGPT, 2023), WildChat (Zhao et al., 2024), and MATH (Hendrycks et al., 2021)). As illustrated in Figure 18, FOCUS demonstrates a substantial performance advantage, achieving up to **8.35× the throughput** of Fast-dLLM v2 on ShareGPT. This significant throughput gap is primarily driven by two critical factors:

- **Lack of System-Level Optimizations:** The Fast-dLLM v2 framework does not currently support production-grade memory management and scheduling techniques, such as *Continuous Batching* (Yu et al., 2022) and *PagedAttention* (Kwon et al., 2023). This severely limits its ability to amortize overheads across concurrent requests compared to the LMDeploy-based architecture of FOCUS.

- **Ineffective Redundancy Reduction:** While Fast-dLLM v2 incorporates an intra-block caching mechanism based on

Fast-dLLM (Wu et al., 2026b), it still necessitates periodic recomputation of the KV states for the entire block. Unlike FOCUS, which actively identifies and evicts non-decodable tokens to reduce the number of processed tokens per step, Fast-dLLM v2 fails to tame the compute-bound nature of the diffusion process effectively.

Because the official-code comparison also reflects framework-level differences, we further implement the Fast-dLLM v2 mechanism inside LMDeploy and compare it against FOCUS under the same settings from Section 5.3. Figure 19 shows that this engine-controlled Fast-dLLM v2 variant improves over the vanilla LMDeploy baseline, confirming that its intra-block mechanism provides a real efficiency benefit. Nevertheless, FOCUS exhibits superior scaling at larger batch sizes, delivering up to **1.91× throughput**. The remaining gap stems from the fact that Fast-dLLM v2 still recomputes the full block before decoding each sub-block; under heavy loads, this repeated dense computation limits its throughput gain, which is consistent with the analysis in the Fast-dLLM v2 paper (Wu et al., 2026a).

It is worth noting that the official Fast-dLLM v2 baseline utilizes a smaller 7B parameter model, yet is significantly outperformed by FOCUS running on the larger SDAR-8B-Chat (Cheng et al., 2025) model. Together with the engine-controlled comparison, these results confirm that taming computational redundancy via token eviction is paramount for scalable DLLM inference.

