# OpenReview forum: "FOCUS: DLLMs Know How to Tame Their Compute Bound"
_ICML.cc/2026/Conference — ICML 2026 regular_

### Official Review · Reviewer_LCcq · 2026-03-10

**Soundness:** 2
**Presentation:** 3
**Significance:** 2
**Originality:** 2
**Overall Recommendation:** 4
**Confidence:** 4

**Summary:**

This paper proposes FOCUS, a training-free inference system for diffusion LLMs under the block-diffusion paradigm. The key observation is that, although computation is parallelized over a token block, only a small subset of tokens is actually decodable at each diffusion step, leading to substantial redundant FLOPs and a compute-bound throughput wall. Building on an empirical correlation between early-layer attention–derived importance and token decodability, FOCUS dynamically predicts and evicts non-decodable tokens on-the-fly, thereby focusing Layer 2+ computation on promising candidates. Implemented on top of the production-grade LMDeploy engine, FOCUS reports up to 3.52× throughput improvement while maintaining or improving generation quality across several benchmarks.

**Compliance With Llm Reviewing Policy:**

Affirmed.

**Final Justification:**

Since the authors will address the previously mentioned issues in the revised paper, and a fair comparison with Fast dLLM v2 has been conducted, all my concerns have been resolved.

**Key Questions For Authors:**

1. FOCUS mainly gains speedup by evicting tokens and reducing FLOPs in Layer 2+. However, prior work such as FlashAttention emphasizes that at this token scale, maintaining contiguous memory access can be more critical than reducing compute, whereas token eviction may introduce gather and irregular memory access [3]. How do the authors view this trade-off?

[3] Dao, Tri, et al. "Flashattention: Fast and memory-efficient exact attention with io-awareness." Advances in neural information processing systems 35 (2022): 16344-16359.

**Limitations:**

yes

**Strengths And Weaknesses:**

# Strength
The paper presents an insight (Figure 3): Layer 0 importance provides almost no separation between decodable and non-decodable tokens, whereas starting from Layer 1 the gap becomes clear—decodable tokens consistently absorb more attention mass and non-decodable ones are suppressed. This observation offers strong empirical support for leveraging early-layer signals for token filtering.

# Weakness
1. In Section 3 “Attention as Decodability Predictor”, the core observation on uneven attention distribution is highly similar to prior dLLM inference optimization works such as Sparse-dLLM and d2Cache [1][2]. FOCUS uses incoming attention as a proxy of token importance and explicitly states “Following SnapKV, we apply MaxPool1D to robustly capture local features.” However, Sparse-dLLM has already leveraged stable sparse attention patterns in dLLMs for token selection with an almost identical SnapKV-style MaxPool aggregation [2]. The authors should explicitly discuss these works in the Related Work section to better delineate the specific contributions and novelty of FOCUS compared to existing methods.

2. In the performance evaluation (e.g., Figure 6), the comparisons start from batch size 32 and mainly emphasize throughput gains in large-batch regimes. In practical dLLM deployment, the effective batch size is often much smaller (e.g., 4/8 or lower). Under such small-batch settings, the batching advantages can be substantially different. Therefore, reporting speedups only for batch ≥ 32 may overestimate the real-world benefit. Furthermore, dLLMs are generally considered challenging to scale for large-batch inference; in many practical scenarios, standard autoregressive LLMs are preferred for high-throughput tasks. If the primary breakthrough of this work lies in enabling efficient large-batch inference for dLLMs, it would be beneficial to include the inference speeds of same-scale autoregressive LLMs under similar conditions as a reference baseline.

3. In Figure 15, the authors compare FOCUS against Fast dLLM-v2, reporting a speedup of approximately 9×. In Appendix F.2, this significant throughput gap is attributed to: (1) system-level optimizations and (2) redundancy reduction. However, this comparison appears inequitable. The paper explicitly describes FOCUS as a highly engineered system implementation ("FOCUS integrates scheduler logic and custom Triton kernels for irregular memory access, spanning 4,000+ lines. Appendix E provides details."), whereas Fast dLLM-v2 utilizes a relatively vanilla implementation. If the core contribution is algorithmic optimization rather than engineering effort, the authors should provide a vanilla version of FOCUS or ensure both methods are evaluated under comparable kernel optimization levels to enable a fairer assessment of the algorithmic gains.

[1] Song, Yuerong, et al. "Sparse-dllm: Accelerating diffusion llms with dynamic cache eviction." arXiv preprint arXiv:2508.02558 (2025).
[2] Jiang, Yuchu, et al. "d$^2$Cache: Accelerating Diffusion-Based LLMs via Dual Adaptive Caching." arXiv preprint arXiv:2509.23094 (2025).

---

> ### Author Rebuttal · Authors · 2026-03-29
>
> Thank you for the careful review and for recognizing the core insight. We respond to your concerns below.
>
> 1. **Relation to prior attention-based methods.**
>
>    We agree that the submission does not sufficiently position FOCUS with respect to other attention-based optimizations. Due to page limits, the current version lacks a discussion section.
>
>    In the revision, we will explicitly cite and discuss prior methods such as **Sparse-dLLM** [1] and **d$^ 2$Cache** [2]. FOCUS differs in operating **target** and regime: we find that the early-layer importance delta correlates with token decodability and use it for **query-token eviction**. Prior approaches use attention signals to **compress or reuse historical KV states**. We will clarify this distinction and discuss that these directions are **complementary**.
>
> 2. **Large-batch emphasis and practical value.**
>
>    FOCUS targets the **high-concurrency** regime, an important point in AI systems. Prior work such as **vLLM** [3] and **DistServe** [4] centers batching and capacity for high-throughput serving. In DLLMs, inference becomes **compute-bound** as batch size grows, and FOCUS mitigates it by evicting redundant tokens.
>
>    We add the **batch-size-1** result (**Table R3** in our response to **Reviewer d53j**). FOCUS has slightly lower throughput than LMDeploy overall, where the computation is too small to benefit from token eviction. It is designed for large batches and can fall back to the LMDeploy path when concurrency is low.
>
>    We agree an ARLLM comparison helps and are actively evaluating this. Currently, a significant scheduling gap between ARLLMs and DLLMs in LMDeploy and FOCUS obscures a fair comparison, highlighting room for system-level optimization. Theoretically, on A100/H100, FOCUS with MoE DLLMs (e.g., LLaDA2.0) can outperform AR counterparts, though Dense models remain compute-bound at large batches. DLLM efficiency can further improve by integrating block-recomputation-free methods like WeDLM [5]. Finally, DLLMs offer bidirectional context and are not merely ARLLM accelerators. If results are available during the discussion, we will report them; otherwise, we will include this comparison in the revision.
>
> 3. **Fairness of the comparison with Fast-dLLM v2.**
>
>    To ensure fairness, we have implemented the Fast-dLLM v2 mechanism inside **LMDeploy**. We compared it against FOCUS under the same settings from Section 5.3.
>
>    **Table R5** shows that while Fast-dLLM v2 improves over the LMDeploy, FOCUS shows superior scaling, delivering up to **1.91×** throughput. Since Fast-dLLM v2 still recomputes the block before decoding each sub-block, its speedup remains modest under heavy loads—consistent with their paper. We will clarify this in the revision.
>
>    **Table R5. Throughput (token/s) comparison between LMDeploy+Fast-dLLM v2 and FOCUS.**
>
>    **SDAR**
>
>    |Dataset|Batch Size|Fast-dLLM v2|FOCUS|
>    |:-|-:|-:|-:|
>    |ShareGPT|32|1019|1218|
>    |ShareGPT|64|1153|1850|
>    |ShareGPT|128|1224|2160|
>    |ShareGPT|256|1253|2272|
>    |WildChat|32|1032|1292|
>    |WildChat|64|1215|1890|
>    |WildChat|128|1338|2205|
>    |WildChat|256|1217|2324|
>    |MATH|32|1413|1766|
>    |MATH|64|1555|2500|
>    |MATH|128|1623|2788|
>    |MATH|256|1612|2930|
>
>    **LLaDA2.0**
>
>    |Dataset|Batch Size|Fast-dLLM v2|FOCUS|
>    |:-|-:|-:|-:|
>    |ShareGPT|32|1051|973|
>    |ShareGPT|64|1774|1644|
>    |ShareGPT|128|2517|2495|
>    |ShareGPT|256|2827|3369|
>    |WildChat|32|952|917|
>    |WildChat|64|1665|1551|
>    |WildChat|128|2403|2393|
>    |WildChat|256|2789|3229|
>    |MATH|32|1711|1544|
>    |MATH|64|2815|2530|
>    |MATH|128|3700|3800|
>    |MATH|256|3916|5017|
>
>
> 4. **Trade-off between FLOPs reduction and irregular memory access.**
>
>    Token eviction is followed by a **compaction** step, after which subsequent layers operate on reduced but **contiguous** tensors (Appendix E.2.3). Thus, the additional memory irregularity occurs only once. Moreover, the irregularity is limited: on the query side, variable query lengths only affect tile scheduling in the ragged attention kernel; on the KV side, FOCUS does not change the layout of the KV cache, so **cache access remains the same** as in the baseline. The benefit extends to all later FFN/MLP computation. Empirically, the scheduler/selection overhead is small relative to full-step latency (~1%, Sec. 5.3), consistent with the observed gains.
>
> **References**
>
> [1] Song, Yuerong, et al. "Sparse-dLLM: Accelerating Diffusion LLMs with Dynamic Cache Eviction." AAAI 2026.
>
> [2] Jiang, Yuchu, et al. "d$^ 2$Cache: Accelerating Diffusion-Based LLMs via Dual Adaptive Caching." ICLR 2026.
>
> [3] Kwon, Woosuk, et al. "Efficient Memory Management for Large Language Model Serving with PagedAttention." SOSP 2023.
>
> [4] Zhong, Yinmin, et al. "DistServe: Disaggregating Prefill and Decoding for Goodput-optimized Large Language Model Serving." OSDI 2024.
>
> [5] Liu, Aiwei, et al. "WeDLM: Reconciling Diffusion Language Models with Standard Causal Attention for Fast Inference." arXiv 2025.

---

> > ### Author Rebuttal · Reviewer_LCcq · 2026-04-03
> >
> > Since the authors will address the previously mentioned issues in the revised paper, and a fair comparison with Fast dLLM v2 has been conducted, all my concerns have been resolved.

---

> > > ### Author Response · Authors · 2026-04-04
> > >
> > > Thank you for reviewing our rebuttal and for your positive acknowledgement. We are glad that your concerns have been resolved. We will revise the paper according to your valuable suggestions.

---

### Official Review · Reviewer_NNs6 · 2026-03-13

**Soundness:** 3
**Presentation:** 3
**Significance:** 3
**Originality:** 3
**Overall Recommendation:** 5
**Confidence:** 3

**Summary:**

When performing block-wise inference on a dLLM, only around ~10% of token are decoded per step. The authors discover that early-layer attention score deltas correlate strongly with token decidability, and propose FOCUS: a training-free system that dynamically evicts “non-decodable” tokens after the first two transformer layers. This reduces FLOPs by 65-80% and achieves up to 3.52x throughput gains over LMDeploy.

**Compliance With Llm Reviewing Policy:**

Affirmed.

**Final Justification:**

FOCUS identifies a genuine and well-validated inefficiency in DLLM inference and provides a practical, training-free solution with substantial throughput gains against a production-grade baseline. The rebuttal addressed the key concerns: H100 results confirm cross-hardware generality and the newly provided Pareto plot demonstrates that FOCUS consistently dominates the baseline across (α, Conf) configurations in both quality and throughput. I recommend acceptance.

**Key Questions For Authors:**

1. In Appendix D.1, can the Chernoff bound be replaced by the tighter Mill’s ratio? ($\frac{1}{x\sqrt{2\pi}}\exp{-x^2 / 2}$)
2. In appendix E3.1, N_decoded is defined as a mean, but in 3.2 is referred to as an exponential moving average. Which one is the correct one?
3. Can you provide a joint quality vs. throughput plot across (alpha, Conf) grid to assess the Pareto optimality?
4. How does FOCUS behave on newer generations of GPU (e.g. H100)?

**Limitations:**

yes

**Strengths And Weaknesses:**

# Strengths

1. **Clear motivation**: Figure 2 quantitatively shows that 90% of block-wise computation is wasted.
2. **Core observation**: the paper demonstrates that the majority of block-wise computation is wasted, with a median of just 1 decoded token per step across benchmarks. The authors then show that the difference between Layr 0 and Layer 1 attention scores is a decodability predictor, and validate this hypothesis empirically as well as giving some insight (Layer 0 lacks cross-token context, whereas Layer 1 already has it).
3. **Thorough system design**: Dynamic budgeting, AR-Context preservation, placeholder integrity and the Neighbor-Aware Stability Criterion collectively address the key failure modes of token eviction. The comparison against LM-Deploy (a production-ready implementation) is the right baseline choice.

# Weaknesses

1. **Quality improvement is not explained**: FOCUS often improves the quality over the baseline (Table 4), attributed to filtering high-confidence but incorrect tokens. No direct evidence supports this: qualitative examples or false-positive rate analysis would support this claim.
2. **Narrow evaluation scope**: the approach is only tested on A100 GPUs. The compute-bound tradeoff differs on newer hardware (H100, B200). It is not clear if this approach will have such high throughput gains is newer generations of accelerators.
3. **Hyperparameter sensitivity without guidance**: Table 4 shows $\alpha$ matters, but no guidance is offered in how to select its value for the downstream task.
4. **No joint quality-throughput tradeoff**: it is currently hard to grasp the trade-off between speedup and task quality. A pareto plot across FOCUS configurations and task accuracy would help understand this tradeoff better.

---

> ### Author Rebuttal · Authors · 2026-03-29
>
> Thank you for the thoughtful and constructive review. We appreciate the positive assessment of our motivation, core observation, and system design, and respond to each concern below.
>
> 1. **Quality improvement is not sufficiently explained.**
>
>    We agree that the current manuscript does not directly isolate the mechanism behind the quality improvements of FOCUS. Our interpretation is that, under relaxed confidence thresholds, the baseline admits more noisy candidates, while FOCUS reduces this negative impact by prioritizing candidates with stronger importance-delta signals.
>
>    At the same time, we would like to clarify why a more controlled analysis is **non-trivial**. In diffusion decoding, defining rigorously whether an intermediate token should be regarded as a positive candidate for eventual decoding is difficult, because this is not directly observable and depends on the evolving denoising trajectory and generated context. This makes it challenging to design a clean controlled experiment that isolates false positives and traces their causal effect on final quality without introducing additional assumptions.
>
>    In the revision, we will therefore tone down this point and present it as a mechanistic interpretation supported by indirect evidence. We will also explicitly note that a deeper controlled analysis of token-level signals and their relationship to final quality is an important direction for future work.
>
> 2. **Narrow evaluation scope / newer GPUs (e.g., H100).**
>
>    We agree that evaluating only on A100 GPUs is a limitation.
>
>    FOCUS reduces processed-token count rather than relying on any A100-specific implementation tricks, so we expect the same qualitative trend to transfer to newer accelerators, although the exact wall-clock gain may vary across hardware.
>
>    To address this concern, we additionally report **H100-80G-PCIe** results under the same setup as Section 5.3 / Figure 6: SDAR and LLaDA2.0 on ShareGPT, WildChat, and MATH, with batch sizes 32–256. These results are summarized in **Table R4** in our response to **Reviewer DGQV**.
>
> 3. **Hyperparameter sensitivity and guidance for choosing $\alpha$.**
>
>    We agree that Table 4 alone does not yet provide enough guidance for choosing $\alpha$.
>
>    Our goal was to show that $\alpha$ is the only method-specific hyperparameter introduced by FOCUS, that performance remains reasonably stable in the range $\{1.2, 1.5, 1.8\}$, and that the default setting $\alpha = 1.5$ with confidence_threshold = 0.8 provides a strong quality/speed tradeoff.
>
>    To make this more explicit, we additionally report throughput under different expansion factors in **Table R1** in our response to **Reviewer d53j**. These results show that throughput remains strong across a wide range of $\alpha$, while $\alpha = 1.5$ remains a balanced default when combined with the quality results in Table 4.
>
> 4. **Missing joint quality-throughput tradeoff view.**
>
>    We appreciate this valuable suggestion. As discussed in the paper, both the confidence threshold and $\alpha$ **simultaneously affect** generation **quality** and **throughput**, since they both change the decoding trajectory and the amount of computation. This is also consistent with the additional throughput results in **Table R1**, where different $\alpha$ values lead to different throughput behaviors under different load conditions.
>
>    At the same time, producing a meaningful joint plot would require evaluating many combinations of confidence thresholds and $\alpha$ values. We are currently running these experiments. If the results become available during the discussion period, we will report them there; otherwise, we will include the joint quality-throughput plot and clarify this tradeoff more systematically in the revision.
>
> 5. **Appendix D: Chernoff bound vs. Mills’ ratio.**
>
>    We thank the reviewer for this insightful suggestion. **Mills’ ratio** indeed provides a **tighter** Gaussian tail bound than the Chernoff-style bound we initially used in Appendix D.2.
>
>    Replacing the Chernoff bound with the Mills’ ratio form $\frac{1}{x\sqrt{2\pi}} \exp(-x^2/2)$ makes the high-SNR decay more explicit, including the additional $\frac{1}{\gamma - 1}$ factor. In the revised manuscript, we will update Proposition D.1 and the corresponding proof accordingly. This will tighten the bound and sharpen the presentation of our theoretical argument.
>
> 6. **Inconsistency on `N_decoded`.**
>
>    Thank you for catching this inconsistency. The current manuscript is not fully consistent in how `N_decoded` is described.
>
>    To clarify, the definition in Appendix E.3.1 is the correct one: `N_decoded` denotes the **cumulative moving average** number of decoded tokens per step for the current request, computed as the arithmetic mean of the decoded-token counts observed so far within that request. It is therefore not an exponential moving average.
>
>    We will revise the manuscript accordingly in Section 4.1, Appendix E.3, and the pseudocode.

---

> > ### Author Rebuttal · Reviewer_NNs6 · 2026-04-03
> >
> > We thank the authors for their replies. My concerns have been addressed. I nonetheless remain interested on the throughput-quality tradeoff results, and hope that the authors can provide some results during the discussion period.

---

> > > ### Author Response · Authors · 2026-04-03
> > >
> > > Thank you for your continued engagement and for following up on the joint quality-throughput tradeoff. We have completed extended measurements to provide a clearer view of this tradeoff. To isolate the tradeoff dynamics, we evaluated both the LMDeploy baseline and our FOCUS system on the SDAR-8B-Chat-b32 model using the GSM8K dataset, with batch size 256. We re-measured generation quality and throughput across confidence thresholds for LMDeploy ($Conf. \in {0.5, 0.55, 0.6, 0.65, 0.7, 0.75, 0.8, 0.85, 0.9}$), and across confidence thresholds and expansion factors for FOCUS ($Conf. \in {0.5, 0.6, 0.7, 0.8, 0.9}$, $\alpha \in {1.2, 1.5, 1.8, 2.1}$). The resulting tradeoff space is illustrated below:
> > >
> > > ```
> > > Quality vs. Throughput Tradeoff (GSM8K)
> > >
> > >  Quality
> > >  (Accuracy %)
> > >    92 |
> > >       |                                       A  B  C   C  B
> > >       |                                            A   D    D AB   C
> > >    88 |          L                                               D
> > >       |                L    L                          A
> > >       |                          L                       B
> > >    84 |                                                            C
> > >       |                                L                           D
> > >       |
> > >    80 |                                 L          A
> > >       |
> > >       |
> > >    76 |                                    L               B       C
> > >       |                                                            D
> > >       |
> > >    72 |
> > >       |
> > >       |                                     L
> > >    68 |
> > >       |
> > >       |
> > >    64 |
> > >       |
> > >       |                                     L
> > >    60 |
> > >       +-----------------------------------------------------------------> Throughput
> > >        1000      1200      1400      1600      1800      2000      2200   (tokens/s)
> > >
> > > Legend:
> > >   L: LMDeploy (Baseline, No Alpha)
> > >   A, B, C, D: FOCUS with α = 1.2, 1.5, 1.8, 2.1
> > > ```
> > >
> > > We highlight two key takeaways from these results:
> > >
> > > 1. **Consistent Pareto Improvement**: Compared to LMDeploy, FOCUS consistently moves the quality-throughput frontier outward over the tested range, yielding a stronger Pareto frontier.
> > > 2. **The "Vertical Wall" and Markov Chain Dynamics**: Interestingly, as we discussed in the paper, DLLM decoding operates like a Markov chain. Simply sacrificing quality—by aggressively lowering the confidence threshold or applying an excessively large $\alpha$—does not guarantee a proportional throughput improvement. Injecting noisy, non-decodable tokens corrupts the subsequent decoding trajectory. This phenomenon is clearly visible as the "vertical wall" on the right side of the plot for both LMDeploy and FOCUS, where a massive loss in quality yields almost zero additional speedup, and in some cases, even a slight slowdown.
> > >
> > > These results reinforce the design of our system: FOCUS successfully tames the compute bound and scales throughput efficiently, while the identified balanced operating point ($\alpha=1.5$, $Conf.=0.8$) navigates the tradeoff space to preserve generation quality. We will include this joint tradeoff analysis in the revised manuscript to provide readers with a more comprehensive view of the system's performance boundaries. We use GSM8K here because it provides a sufficient sample size to clearly observe these granular tradeoff dynamics. As additional validation, we will also examine this joint tradeoff on other datasets, such as MBPP, in future work. Thank you again for driving this insightful and valuable discussion!

---

### Official Review · Reviewer_DGQV · 2026-03-13

**Soundness:** 4
**Presentation:** 3
**Significance:** 4
**Originality:** 4
**Overall Recommendation:** 5
**Confidence:** 4

**Summary:**

This paper proposes FOCUS, a training-free inference system for block diffusion LLMs that improves throughput by predicting decodable tokens early and evicting non-decodable tokens on the fly. The key insight is that early-layer attention importance, especially the importance delta between Layer 0 and Layer 1, is strongly correlated with token decodability. Based on this, the paper introduces a dynamic token budgeting and eviction strategy to reduce redundant computation in compute-bound DLLM decoding. The empirical results show up to 2.32× throughput improvement at standard block size and 3.52× at larger block size on fast LLM inference framework LMDeploy, while preserving or even improving generation quality across multiple benchmarks.

**Compliance With Llm Reviewing Policy:**

Affirmed.

**Final Justification:**

FOCUS addresses a practical and important inefficiency in block diffusion LLM inference through a clean, well-motivated design. Using early-layer attention importance-delta as a decodability predictor is novel, and the resulting token eviction framework delivers strong throughput gains while preserving generation quality.

My main concern about evaluation on a single GPU has been fully resolved. The new H100 experiments confirm that FOCUS maintains meaningful speedups on newer hardware, with up to 2.33× throughput improvement, though exact gains vary by hardware-model-workload combination as expected. The discussion on compatibility with token editing was also reasonable and honest.

I keep my score at 5 (accept). The paper is technically strong, tackles a meaningful problem, and delivers a practical, well-engineered contribution with solid evaluation across models, datasets, and now multiple hardware platforms.

**Key Questions For Authors:**

1. Some dLLMs now support token editing. Could you discuss how FOCUS would work with token editing?
2. Could you evaluate FOCUS on different GPUs to see how the speedup changes?

**Limitations:**

Yes.

**Strengths And Weaknesses:**

**Strengths**
1. Novel and well-motivated design.
The paper identifies an annoying inefficiency in block diffusion inference: full-block computation is repeated each step even though only a small fraction of tokens are decodable. Using early-layer attention as a decodability predictor is an interesting and well-motivated idea, and the resulting token eviction framework is clean.
2. High practical value.
dLLM is well-known only useful in low batch size setting because large block size will make the forward pass very expensive in the compute-bound large batch size setting. This work tackles this issue, which is an important direction and has high practical value.
3. Strong empirical results.
The evaluation is solid. The paper reports substantial throughput gains over LMDeploy, including 2.32× throughput improvement at standard settings and 3.52× at larger block sizes, while preserving or improving generation quality. The paper also evaluates on both dense SDAR and MoE LLaDA2.0 models, which strengthens the practical relevance.
4. Good ablations and implementation effort.
The paper does a good job validating its core design choices, including the importance-based token selection and the neighbor-aware cache strategy. The system is also implemented on LMDeploy, which improves reproducibility and practical credibility.

**Weakness**
1. Lack of evaluation on various hardware settings.
The evaluation is purely conducted on single A100 GPU. However, different GPU like customer GPU or later generation NVIDIA GPU like Hopper and Blackwell GPUs will have different FLOPS/Bandwidth ratio, which will affect the speedup.

---

> ### Author Rebuttal · Authors · 2026-03-29
>
> Thank you for the constructive feedback and for the positive assessment of the novelty, practical relevance, and empirical strength of our work. We address the suggestions point by point below.
>
> 1. **How would FOCUS work with token editing?**
>
> 	This is a very interesting question. Recent DLLMs such as **LLaDA2.1** [1] show that token editing can increase decoding parallelism while maintaining competitive generation quality. However, token editing still typically requires computing the logits of all tokens at each step in order to determine which positions should be edited or decoded.
>
> 	We believe FOCUS is highly **complementary** to this direction. The core idea of FOCUS is to use early-layer attention importance-delta signals to predict which tokens are likely to be decodable at the current step. In a token-editing setting, the same idea could potentially be extended to predict which positions are likely to require recomputation for editing at the current step, instead of uniformly computing token logits for all positions at every iteration.
>
> 	We therefore believe the central insight of FOCUS may also help reduce the redundant computation that remains in current token-editing-based DLLM inference. We think this is a promising and important future direction, and we will explicitly discuss it in the revision.
>
> 2. **Could you evaluate FOCUS on different GPUs to see how the speedup changes?**
>
> 	We agree that evaluating only on a single A100 is a limitation, since realized wall-clock speedup can depend on the target GPU’s compute/memory characteristics. FOCUS primarily improves efficiency by reducing the number of processed tokens and the resulting compute in later layers, rather than relying on A100-specific implementation tricks. We therefore expect its qualitative advantage to persist on newer GPUs, although the exact speedup may vary across architectures.
>
> 	To directly address this concern, we additionally ran experiments on **H100-80G-PCIe** under the same throughput setting as Section 5.3 and Figure 6: SDAR and LLaDA2.0 on ShareGPT, WildChat, and MATH, with batch sizes 32, 64, 128, and 256. We report the corresponding H100 results below to assess whether FOCUS maintains its advantage over LMDeploy on the newer Hopper architecture, delivering up to **2.33×** throughput improvement.
>
> 	**Table R4. H100 throughput (token/s) comparison between LMDeploy and FOCUS.**
>
> 	**SDAR**
>
> 	|Dataset|Batch Size|LMDeploy|FOCUS|Speedup|
> 	|:-|-:|-:|-:|-:|
> 	|ShareGPT|32|1433|1801|1.26×|
> 	|ShareGPT|64|1507|2813|1.87×|
> 	|ShareGPT|128|1553|3572|2.30×|
> 	|ShareGPT|256|1593|3718|2.33×|
> 	|WildChat|32|1533|1935|1.26×|
> 	|WildChat|64|1593|2921|1.83×|
> 	|WildChat|128|1626|3555|2.19×|
> 	|WildChat|256|1666|3705|2.22×|
> 	|MATH|32|2794|2689|0.96×|
> 	|MATH|64|2860|4126|1.44×|
> 	|MATH|128|2882|4861|1.69×|
> 	|MATH|256|2950|4894|1.66×|
>
> 	**LLaDA2.0**
>
> 	|Dataset|Batch Size|LMDeploy|FOCUS|Speedup|
> 	|:-|-:|-:|-:|-:|
> 	|ShareGPT|32|640|1387|2.17×|
> 	|ShareGPT|64|2322|2345|1.01×|
> 	|ShareGPT|128|2831|3510|1.24×|
> 	|ShareGPT|256|2888|4457|1.54×|
> 	|WildChat|32|643|1314|2.04×|
> 	|WildChat|64|2100|2228|1.06×|
> 	|WildChat|128|2606|3379|1.30×|
> 	|WildChat|256|2616|4319|1.65×|
> 	|MATH|32|1065|2111|1.98×|
> 	|MATH|64|4272|3501|0.82×|
> 	|MATH|128|5125|5208|1.02×|
> 	|MATH|256|5118|6320|1.23×|
>
>    The results suggest that the exact realized speedup indeed depends on the hardware-model-workload combination. We will add these H100 results and this discussion in the revision for completeness.
>
> **Reference**
>
> [1] Bie, Tiwei, et al. "LLaDA2.1: Speeding Up Text Diffusion via Token Editing." *arXiv preprint arXiv:2602.08676* (2026).

---

> > ### Author Rebuttal · Reviewer_DGQV · 2026-04-03
> >
> > I appreciate the authors’ detailed response and additional experiments. My main questions were addressed, and I will keep my positive rating. Good work.

---

> > > ### Author Response · Authors · 2026-04-04
> > >
> > > Thank you for your time and the positive evaluation. We will incorporate the additional discussion and experiments into the revised paper, and sincerely appreciate your support.

---

### Official Review · Reviewer_d53j · 2026-03-18

**Soundness:** 4
**Presentation:** 4
**Significance:** 4
**Originality:** 3
**Overall Recommendation:** 6
**Confidence:** 4

**Summary:**

While Diffusion LLMs (DLLMs) tend to be faster than Autoregressive LLMs (ARLLMs), they suffer from inefficiency in running many forward FLOPs in return for few decoded tokens per forward pass.
This paper proposes to solve this problem by:
a) using attention scores to determine which tokens are decodable, and hence evicts tokens that are unlikely to be decodable (there are more details in the algorithm such as that all preceding tokens to a token that is expected to be decodable are not evicted)
b) caching KVs of tokens within a block from left to right when multiple conditions are met

Results lead to upto 3.52x speedups while maintaining accuracy.

**Compliance With Llm Reviewing Policy:**

Affirmed.

**Final Justification:**

I would like to keep my Strong Accept score. The authors addressed my concerns. I skimmed through the other reviews and they seem to be positive as well.

**Key Questions For Authors:**

- Table 2, the $O$ complexity symbol in DLLM column is boldened while it is not boldened in the ARLLM column
- Can you also try natural language tasks, such as Natural Questions, or TriviaQA.
- Line 147: Please clarify whether $S_{i,j} = Q_{i}K_{j}^{T} / sqrt(d_{k}) $. Or in layman terms, whether $S_{i,j}$ is the score of the ith query with jth key
- No results for batch size 1 were provided. Even if there is no speedup on batch size 1, I believe it should be provided. If it doesn't lead to speedup, it could be listed as a limitation.

**Limitations:**

No limitations section was provided. I don't have limitations to mention at the top of my head.

**Strengths And Weaknesses:**

Strengths:
- Proposed solution is backed by intuition and analysis
- Significant speedups (upto 3.52x) while maintaining accuracy
- Maintains accuracy when confidence threshold is reduced (to me that was impressive and I believe that is significant in the DLLM research field)
- Evaluated speeds on natural language datasets (such as ShareGPT and WildChat) while most DLLMs evaluate speed on datasets like GSM8K (that tend to lead to inflated speedups because of the low underlying uncertainty of math datasets)
- Approach is training free
- Approach introduces a single hyperparameter, that seems to be easy to tune
- Implemented solution on top of a rigorous baseline, LMDeploy, and showed that speedups obtained on top of existing optimizations in the field
- Impressive reduction in FLOPs redundancy shown in Table 5 (up to 79% reduction)

Weaknesses:
- Did not present a accuracy-vs-.speed plot comparing different diffusion and AR models/approaches

---

> ### Author Rebuttal · Authors · 2026-03-30
>
> Thank you for the very positive and careful review. We are glad that the reviewer found the intuition, analysis and results convincing. We address the suggestions below.
>
> 1. **Accuracy-speed tradeoff.**
>
>    We agree that a unified accuracy-vs-speed plot across approaches would be valuable. However, such a comparison is **not straightforward**, because these methods differ in paradigm. Some are not yet readily supported in engine. As a result, a controlled comparison would require substantial work to ensure fairness. We view this as important future work.
>
>    Following your suggestion, we made the tradeoff more explicit for FOCUS. In the paper, Table 4 reports quality across confidence thresholds and $\alpha$, while the throughput section focuses on the default configuration. We additionally measured throughput for $\alpha \in \{1.2, 1.8\}$ under the same settings from Section 5.3, and report the results below with the $\alpha=1.5$.
>
>    **Table R1. Throughput (tokens/s) under different $\alpha$.**
>
>    **SDAR**
>
>    |Dataset|Batch Size|$\alpha=1.2$|$\alpha=1.5$|$\alpha=1.8$|
>    |:-|-:|-:|-:|-:|
>    |ShareGPT|32|1162|1218|1270|
>    |ShareGPT|64|1737|1850|1887|
>    |ShareGPT|128|2038|2160|2206|
>    |ShareGPT|256|2213|2272|2295|
>    |WildChat|32|1201|1292|1361|
>    |WildChat|64|1786|1890|1932|
>    |WildChat|128|2096|2205|2212|
>    |WildChat|256|2218|2324|2344|
>    |MATH|32|1638|1766|1914|
>    |MATH|64|2427|2500|2732|
>    |MATH|128|2803|2788|2995|
>    |MATH|256|2925|2930|3126|
>
>    **LLaDA2.0**
>
>    |Dataset|Batch Size|$\alpha=1.2$|$\alpha=1.5$|$\alpha=1.8$|
>    |:-|-:|-:|-:|-:|
>    |ShareGPT|32|948|973|996|
>    |ShareGPT|64|1603|1644|1685|
>    |ShareGPT|128|2414|2495|2527|
>    |ShareGPT|256|3267|3369|3390|
>    |WildChat|32|907|917|952|
>    |WildChat|64|1539|1551|1607|
>    |WildChat|128|2339|2393|2425|
>    |WildChat|256|3149|3229|3239|
>    |MATH|32|1467|1544|1651|
>    |MATH|64|2413|2530|2666|
>    |MATH|128|3549|3800|3950|
>    |MATH|256|4785|5017|4758|
>
>    In general, larger $\alpha$ improves throughput by retaining more potential tokens, especially at small batch sizes. The trend is not monotonic in every setting, since aggressive expansion can introduce redundant computation. Combined with Table 4, these measurements support our choice of $\alpha=1.5$ as a balanced default. More broadly, throughput **remains strong** across a wide range of $\alpha$. We will include these results in the revision.
>
> 2. **Table 2 formatting.**
>
>    We will fix the boldface inconsistency in the complexity notation.
>
> 3. **Additional natural-language tasks.**
>
>    We have now evaluated FOCUS on two QA-style benchmarks, **Natural Questions (NQ-Open)** and **TriviaQA**, using the same grid as Table 4. The results are summarized below.
>
>    **Table R2. Natural-language QA results across thresholds and $\alpha$.**
>
>    **SDAR**
>
>    |Conf.|M|$\alpha$|NQ-Open|TriviaQA|
>    |-:|:-:|:-:|-:|-:|
>    |0.9|B|-|21.86|55.75|
>    |0.9|F|1.2|22.44|57.24|
>    |0.9|F|1.5|22.41|57.24|
>    |0.9|F|1.8|22.24|57.20|
>    |0.8|B|-|21.61|55.37|
>    |0.8|F|1.2|22.08|56.89|
>    |0.8|F|1.5|21.91|56.87|
>    |0.8|F|1.8|21.63|56.68|
>    |0.7|B|-|21.14|54.45|
>    |0.7|F|1.2|21.97|56.17|
>    |0.7|F|1.5|21.27|55.97|
>    |0.7|F|1.8|21.47|56.14|
>
>    **LLaDA2.0**
>
>    |Conf.|M|$\alpha$|NQ-Open|TriviaQA|
>    |-:|:-:|:-:|-:|-:|
>    |0.9|B|-|17.59|51.33|
>    |0.9|F|1.2|17.76|52.69|
>    |0.9|F|1.5|17.70|52.66|
>    |0.9|F|1.8|17.42|52.56|
>    |0.8|B|-|16.90|50.27|
>    |0.8|F|1.2|17.45|51.73|
>    |0.8|F|1.5|17.29|51.60|
>    |0.8|F|1.8|17.26|51.57|
>    |0.7|B|-|16.54|48.91|
>    |0.7|F|1.2|17.01|50.24|
>    |0.7|F|1.5|17.12|50.21|
>    |0.7|F|1.8|16.79|50.20|
>
>    These results are **consistent** with our main findings: FOCUS generally maintains or improves quality on additional natural-language QA tasks.
>
> 4. **Clarification of Eq. 2.**
>
>    Yes. We will clarify that $S^{(h)}_{i,j} = Q_i K_j^\top / \sqrt{d_k}$, i.e., the pre-softmax attention score between the $i$-th query and the $j$-th key in head $h$, with $i,j$ indexing token positions within the current block.
>
> 5. **Response to the batch-size-1 question.**
>
>    We agree that the batch-size-1 setting should be reported explicitly. FOCUS targets the high-concurrency, compute-bound regime, where it improves throughput scaling by evicting non-decodable tokens. At batch size 1, the computation is too small for FOCUS to improve throughput, and scheduler overhead becomes more visible. In our additional BS=1 measurement, FOCUS is slightly below the LMDeploy on average.
>
>    **Table R3. Throughput (token/s) at batch size 1.**
>
>    |Model|Dataset|LMDeploy|FOCUS|
>    |:-|:-|-:|-:|
>    |SDAR|ShareGPT|57.5|63.3|
>    |SDAR|WildChat|57.2|63.2|
>    |SDAR|MATH|122.5|89.6|
>    |LLaDA2.0|ShareGPT|45.5|43.2|
>    |LLaDA2.0|WildChat|42.6|41.7|
>    |LLaDA2.0|MATH|90.7|70.9|
>
>    As a practical note, FOCUS can be enabled only when the running batch is sufficiently large, while very small batches fall back to the original baseline decoding path. We will add this guideline to the discussion section in the revision.

---

> > ### Author Rebuttal · Reviewer_d53j · 2026-04-04
> >
> > Thanks for addressing my concern. Please add the results for batch size 1 and mention them in the Limitations section, and emphasize more in the Abstract that the solution speeds up batch size > 1.

---

> > > ### Author Response · Authors · 2026-04-04
> > >
> > > Thank you for your time and the positive feedback. We deeply appreciate your constructive suggestions, and will improve our paper accordingly, including updating the Limitations and Abstract.

---

### Decision · Program_Chairs · 2026-04-30

**Decision:**

Accept (regular)

**Comment:**

This paper identifies a key inference inefficiency in diffusion LLMs: although decoding is parallelized over token blocks, only a small subset of tokens is decodable at each step, resulting in substantial wasted computation. To address this issue, the authors propose FOCUS, which dynamically prioritizes likely decodable tokens and evicts non-decodable ones during inference, thereby improving effective batch utilization. Experiments demonstrate up to 3.52× higher throughput than LMDeploy while maintaining or improving generation quality.

Overall, the reviewers agree that this paper makes solid and novel contributions. I therefore recommend accepting this paper.